# TIME-IMM: A Dataset and Benchmark for Irregular Multimodal Multivariate Time Series

**Ching Chang**[1,2*]   **Jeehyun Hwang**[1]   **Yidan Shi**[1]   **Haixin Wang**[1]
**Wen-Chih Peng**[2]   **Tien-Fu Chen**[2]   **Wei Wang**[1]
[1]University of California, Los Angeles   [2]National Yang Ming Chiao Tung University

## Abstract

Time series data in real-world applications such as healthcare, climate modeling, and finance are often irregular, multimodal, and messy, with varying sampling rates, asynchronous modalities, and pervasive missingness. However, existing benchmarks typically assume clean, regularly sampled, unimodal data, creating a significant gap between research and real-world deployment. We introduce TIME-IMM, a dataset specifically designed to capture cause-driven irregularity in multimodal multivariate time series. TIME-IMM represents nine distinct types of time series irregularity, categorized into trigger-based, constraint-based, and artifact-based mechanisms. Complementing the dataset, we introduce IMM-TSF, a benchmark library for forecasting on irregular multimodal time series, enabling asynchronous integration and realistic evaluation. IMM-TSF includes specialized fusion modules, including a timestamp-to-text fusion module and a multimodality fusion module, which support both recency-aware averaging and attention-based integration strategies. Empirical results demonstrate that explicitly modeling multimodality on irregular time series data leads to substantial gains in forecasting performance. TIME-IMM and IMM-TSF provide a foundation for advancing time series analysis under real-world conditions. The dataset is publicly available at `https://github.com/blacksnail789521/Time-IMM`, and the benchmark library can be accessed at `https://github.com/blacksnail789521/IMM-TSF`.

## 1   Introduction

Time series analysis plays a foundational role across diverse fields such as healthcare [40, 54], climate science [56, 10], and finance [18, 81]. It underpins critical applications such as patient health monitoring, weather forecasting, disaster response coordination, and financial trend prediction. These applications increasingly require modeling time series data that exhibit irregular sampling, asynchronous observation patterns, and pervasive missingness, often across multiple modalities. However, traditional time series research has largely relied on clean, unimodal, regularly-sampled datasets where observations arrive at fixed intervals. These assumptions rarely hold in real-world scenarios. In practice, time series data are often *irregular*, *multimodal*, and *messy*, with variable sampling rates, asynchronous modalities, and pervasive missingness.

Despite the importance of addressing such irregularities, most publicly available time series benchmarks largely overlook these complexities. Datasets such as the UCR archive [14], M4 [44], and many multimodal datasets like Time-MMD [41] assume regular temporal grids and fixed sampling intervals. They do not fully capture the diverse irregularities arising in operational systems, human behavior, and natural processes. This simplification limits the development and evaluation of models capable of handling the challenges faced in real-world deployments.

---

*Correspondence to: Ching Chang <chingchang0730@ucla.edu>

39th Conference on Neural Information Processing Systems (NeurIPS 2025) Track on Datasets and Benchmarks.

The primary obstacle to advancing time series analysis under real-world conditions lies in the absence of benchmarks that accurately reflect the irregular, heterogeneous nature of practical data. Despite the growing body of research on both time series forecasting and multimodal learning, three critical challenges remain: **(1) The oversight of real-world irregularity in existing time series benchmarks**, as most prior datasets [41, 44, 14] prioritize clean, regularly-sampled settings and fail to capture the pragmatic complexities of operational systems and natural processes; **(2) The limited exploration of multimodal integration within irregular time series modeling**, where existing work [78, 56, 57, 6, 58] primarily focuses on unimodal numerical sequences and neglects the asynchronous nature of textual and other auxiliary modalities; and **(3) The missing systematic understanding of irregularity causes**, as prior efforts rarely distinguish between the major categories of irregularity — such as trigger-based events, constraint-based sampling restrictions, and artifact-based system failures — that underlie observed irregular patterns, limiting causal interpretability and model robustness.

To address these challenges, we introduce a new dataset and benchmark library designed for systematic study of irregular multimodal multivariate time series analysis. Our main contributions are:

- **Cause-Driven Irregular Multimodal Dataset.** We construct TIME-IMM, the first benchmark that not only explicitly categorizes real-world irregularities into nine cause-driven types—spanning trigger-based, constraint-based, and artifact-based mechanisms—but also incorporates richly annotated textual data alongside numerical observations. The textual modality captures asynchronous, auxiliary information such as clinical notes, sensor logs, or event descriptions, providing critical context for interpreting and forecasting multivariate time series. This enables comprehensive modeling of complex, cross-modal interactions and irregular sampling behaviors across multiple domains.

- **Benchmark Library for Irregular Multimodal Time Series Forecasting.** We develop IMM-TSF, a plug-and-play benchmarking library for forecasting on irregular multimodal time series. It supports asynchronous integration of numerical and textual data through modular components for encoding and fusion, enabling flexible and realistic experimentation.

- **Empirical Validation of Irregularity and Multimodality Modeling.** Through extensive experiments across diverse types and causes of irregularity, we show that incorporating multimodality on top of irregular time series modeling yields substantial forecasting improvements—achieving an average MSE reduction of 6.71%, and up to 38.38% in datasets with highly informative textual signals. These results underscore the superiority of multimodal approaches in handling the complexities of real-world time series forecasting.

By introducing TIME-IMM and IMM-TSF, we aim to catalyze research in robust, generalizable time series analysis, moving the field beyond simplified assumptions toward truly real-world scenarios. Further details on related work and dataset limitations are provided in Appendix A and Appendix B, respectively.

## 2 TIME-IMM: A Dataset for Irregular Multimodal Multivariate Time Series

While prior benchmarks in time series forecasting and multimodal learning have expanded the diversity of application domains, they largely overlook the intrinsic irregularities that characterize real-world data collection. Existing datasets predominantly assume regular sampling patterns, simple temporal alignments, and domain-driven organization, abstracting away the diverse causes that drive irregular observations [31, 12, 63]. Even recent multimodal datasets [41] focus on domain variety rather than addressing the deeper methodological challenge of irregularity. As a result, current benchmarks fail to capture the pragmatic complexities introduced by trigger-based events, constraint-driven sampling, and artifact-induced disruptions. To bridge this gap, TIME-IMM explicitly centers irregularity as a core characteristic of time series data, categorizing its causes and curating multimodal datasets that reflect real-world irregular structures.

### 2.1 Taxonomy of Irregularity in Time Series

Irregularities in real-world time series are driven by diverse and systematic causes, not just random or arbitrary occurrences. Understanding the origins of these irregularities is crucial for developing models that are robust to realistic conditions and capable of meaningful interpretation. To structure

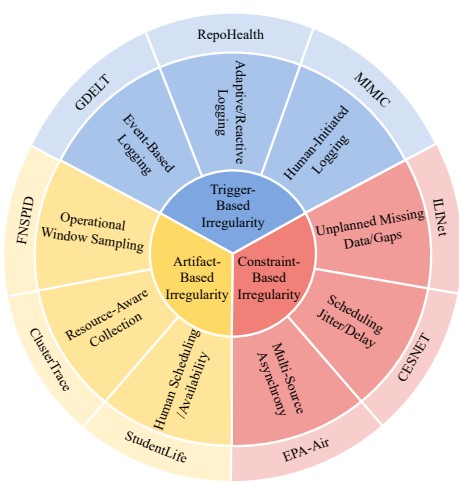

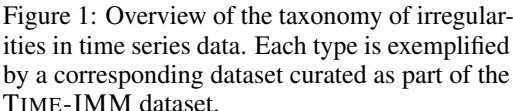

Figure 1: Overview of the taxonomy of irregularities in time series data. Each type is exemplified by a corresponding dataset curated as part of the TIME-IMM dataset.

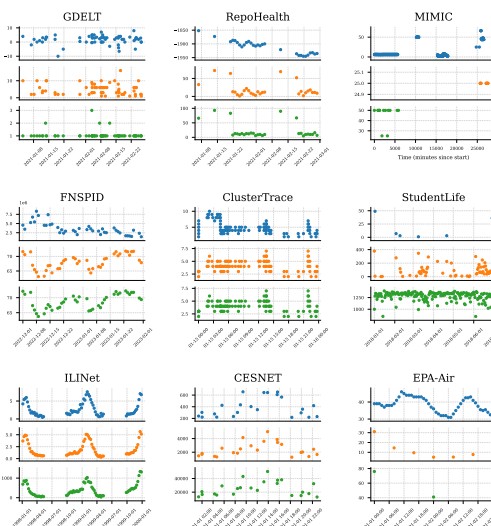

Figure 2: Visualization of the TIME-IMM datasets, annotated to show sampling patterns linked to the taxonomy, such as adaptive bursts (RepoHealth), trading-hour gaps (FNSPID), and sensor misalignment (EPA-Air).

this complexity, we propose a taxonomy based on the primary nature of the factors causing irregularity. An overview of our taxonomy is illustrated in Figure 1.

First, **Trigger-Based Irregularities** arise when data collection is initiated by external events or internal system triggers, meaning observations occur only in response to specific happenings rather than at predetermined intervals. Second, **Constraint-Based Irregularities** are driven by limitations on when data can be collected, such as operational schedules, resource availability, or human factors, where the sampling process is shaped by external restrictions rather than an ideal or intended schedule. Third, **Artifact-Based Irregularities** result from imperfections in the data collection process itself, including failures, delays, and asynchrony, where the intended regularity is disrupted by technical or systemic artifacts.

This taxonomy highlights that real-world irregularities are not caused by a single factor but arise from fundamentally different sources, each requiring tailored modeling strategies to capture their structure and implications. Each dataset in TIME-IMM exemplifies a distinct type of irregularity, as shown in Figure 1. To complement the taxonomy, Figure 2 presents representative visualizations of three features per dataset, illustrating the diversity of sampling patterns and irregular behaviors. Table 1 summarizes dataset statistics in TIME-IMM. We also introduce a set of metrics to characterize time series irregularity—feature observability entropy, temporal observability entropy, and mean inter-observation interval—which respectively quantify: (1) the variability of missingness across features via the normalized Shannon entropy of the feature-presence distribution, (2) the dispersion of events over time using the normalized Shannon entropy of the event-time distribution, and (3) the average time gap between successive observations. Formal definitions of these metrics are provided in Appendix C.

### 2.1.1 Trigger-Based Irregularities

Trigger-based irregularities are characterized by observations that occur only in response to discrete events or system triggers. We further categorize this group into three representative types, each illustrating distinct mechanisms behind event-driven sampling.

**Event-Based Logging** occurs when observations are recorded only upon notable external events, such as geopolitical protests or seismic activity. We illustrate this pattern using the GDELT Global Database [36], where each record corresponds to a discrete global event like a political crisis or major policy change.

Table 1: Overview of datasets in Time-IMM. Columns marked with † refer to the textual modality.

| Dataset | # Entities | # Features | # Unique Timestamps | # Observations | Feature Observability Entropy | Temporal Observability Entropy | Mean Inter-Obseration Interval | # Text Entries† | Textual Temporal Observability Entropy† |
|---|---|---|---|---|---|---|---|---|---|
| GDELT | 8 | 5 | 34317 | 193205 | 1 | 0.9964 | 7.2364 hours | 14357 | 0.9896 |
| RepoHealth | 4 | 10 | 6783 | 67830 | 1 | 0.9658 | 1.8217 days | 12310 | 0.9821 |
| MIMIC | 20 | 30 | 91098 | 219949 | 0.8461 | 0.6556 | 14.6157 minutes | 1593 | 0.6758 |
| FNSPID | 10 | 6 | 3659 | 209688 | 1 | 0.9969 | 1.4507 days | 20826 | 0.9488 |
| ClusterTrace | 3 | 11 | 12615 | 69001 | 0.893 | 0.9753 | 18.1425 minutes | 688 | 0.9971 |
| StudentLife | 20 | 9 | 1743 | 153610 | 0.92 | 0.9775 | 1.0191 days | 6623 | 0.9761 |
| ILINet | 1 | 11 | 4918 | 4918 | 0.9267 | 1 | 6.989 days | 650 | 1 |
| CESNET | 30 | 10 | 51107 | 512760 | 1 | 1 | 1.17 hours | 224 | 0.9869 |
| EPA-Air | 8 | 4 | 6587 | 49552 | 0.3777 | 0.9576 | 1.0242 hours | 1244 | 0.9956 |

**Adaptive or Reactive Sampling** arises when a system automatically adjusts its sampling rate based on observed dynamics, such as a heart monitor increasing frequency during arrhythmia or a network logger reacting to high congestion. Here, the triggering is *system-driven*. In our benchmark, the RepoHealth dataset reflects this behavior, with GitHub project histories sampled more densely during periods of high commit activity and sparsely during quiet phases.

**Human-Initiated Observations** occur when data are recorded by deliberate human action rather than automation—for instance, clinicians measuring vitals only when clinically necessary or users self-reporting symptoms upon noticeable changes. This type is *human-driven*, distinguishing it from the automatic reactivity of adaptive sampling. We represent it using the MIMIC clinical database [27, 28], where vital signs are logged irregularly according to clinicians' judgment.

### 2.1.2 Constraint-Based Irregularities

Constraint-based irregularities occur when external limitations such as operational schedules, resource restrictions, or human factors determine when data can be collected. While all share the characteristic of externally imposed constraints, they differ in their *predictability* and therefore require distinct modeling strategies. We categorize them into three subtypes reflecting this spectrum from highly predictable to human-driven variability.

**Operational Window Sampling** is the most predictable form, where data collection is limited to fixed time windows (for example, trading hours or orbital passes). These patterns follow rigid schedules and can often be modeled using periodic masks or known calendars. We illustrate this type using FNSPID [18], where financial time series are sampled only during market operating hours, creating regular, clock-based gaps.

**Resource-Aware Collection** represents a partly predictable case, where sampling adapts to system resource conditions such as power, bandwidth, or workload. The resulting gaps are conditionally triggered but can often be anticipated using auxiliary signals (for example, device status). This is exemplified by ClusterTrace [68], where GPU cluster telemetry is recorded only when containers are active, yielding structured but dynamic irregularity.

**Human Scheduling / Availability** is the least predictable form, driven by human activity patterns such as staff shifts, sleep cycles, or daily routines. These soft, behavior-dependent gaps benefit from time-aware modeling using cues like time of day or day of week. We illustrate this with StudentLife [46], where smartphone sensing data exhibits irregularity shaped by human schedules and activity cycles.

### 2.1.3 Artifact-Based Irregularities

Artifact-based irregularities arise when technical imperfections, failures, or system noise disrupt the intended regularity of data collection, creating unintended gaps or distortions in the observation sequence. We further categorize this group into three representative types that reflect different sources of system-induced irregularity.

**Unplanned Missing Data / Gaps** occur when expected data points are unintentionally lost due to accidents, system failures, or transmission errors, leaving empty time steps in the sequence. Unlike

missingness in other categories such as adaptive sampling, human-initiated logging, or multi-source asynchrony, which is intentional or structural, these gaps arise without design or decision. They are typically treated as random missingness (MCAR or MAR) and handled with imputation, whereas intentional gaps are often informative (MNAR) and can carry modeling value. We illustrate this form of irregularity using the ILINet dataset [41] from the CDC, which provides weekly influenza-like illness reports across the United States. Some weeks contain missing or partial data caused by non-reporting providers or technical failures during data transmission.

**Scheduling Jitter / Delay** happens when data points are recorded at uneven time intervals due to system delays or timing issues. Instead of arriving on a regular schedule, the data shows random gaps and timing shifts. This can be caused by logging processes competing for resources, background maintenance tasks, or data throttling during peak hours. We use the CESNET dataset [32] to show this kind of irregularity since its network flow records often appear at inconsistent times due to internal scheduling and system constraints.

**Multi-Source Asynchrony** arises when observations from multiple data streams are integrated despite differing internal clocks, sampling rates, or synchronization policies. Examples include environmental sensor networks where temperature, humidity, and air quality sensors report measurements at unsynchronized intervals, and health monitoring systems where wearable devices and clinical records log data at different temporal granularities. To represent this form of irregularity, we compile datasets from EPA Outdoor Air Sensors [61], where heterogeneous sensors report at varying frequencies, creating inherent asynchrony across measurement streams.

## 2.2 Dataset Construction

To construct TIME-IMM, we start with real-world time series data exhibiting diverse forms of irregularity, then follow a two-stage pipeline: curating relevant textual data to pair with each time series, and preserving asynchronous timestamps across both modalities to reflect real-world conditions.

**Multimodal Textual Data Curation.** Each time series in TIME-IMM is paired with contextual textual information, such as situational narratives, observational reports, or human-generated logs, depending on the nature of the dataset. Since the modality and format of text differ across sources, we adopt dataset-specific strategies for acquisition and preprocessing. Details on data sources and collection pipelines for each dataset are provided in Appendix D.

To ensure the textual modality is both relevant and informative, we apply a preprocessing strategy inspired by Time-MMD [41]. This includes an initial filtering step to retain only semantically relevant documents, followed by summarization to improve clarity and compress dense information into concise, interpretable content. Summarization and filtering are performed jointly using GPT-4.1 Nano [50] with domain-specific prompts that generate a five-sentence summary only if the input is topically relevant, otherwise skipping the document entirely. Details on how semantically relevant documents are identified, along with the full prompt templates and filtering logic, are provided in Appendix E, and representative preprocessing examples are shown in Appendix F. We use summarized textual data instead of original documents because many source datasets are proprietary and cannot be redistributed under the NeurIPS CC BY 4.0 requirement, following the same practice as Time-MMD [41].

**Handling Asynchronous Timestamps.** Unlike traditional multimodal datasets [41, 79] that assume or enforce temporal alignment between modalities, TIME-IMM intentionally preserves asynchronous timestamping. Numerical and textual observations occur on independent, irregular schedules, reflecting real-world conditions where continuous sensors and episodic reports are generated through different processes. Rather than forcing alignment, which can be ambiguous or misleading under irregular sampling, our design allows models to learn soft temporal associations through the timestamp-to-text fusion module described in Section 3.2. This approach retains full flexibility for downstream work to apply alternative alignment strategies, such as window-based or delay-aware matching, without constraining the data itself.

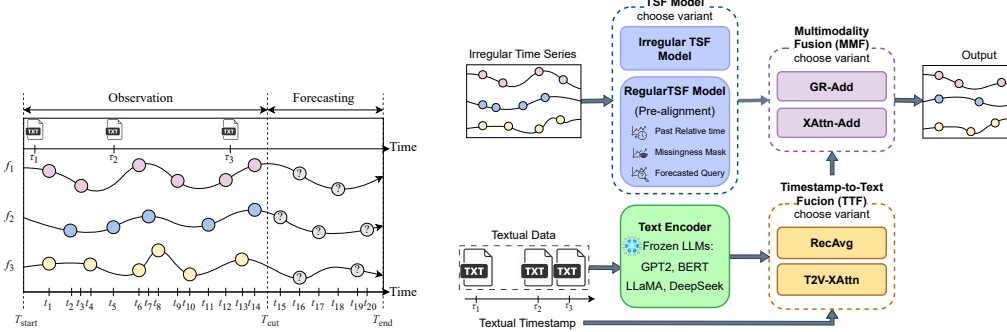

Figure 3: Problem formulation for irregular multimodal multivariate time series forecasting.

Figure 4: IMM-TSF architecture. The library includes modular fusion layers that combine irregular numerical sequences with asynchronous text via timestamp-to-text fusion and multimodal forecast fusion.

## 2.3 Ethical Considerations and Data Access

All datasets in TIME-IMM are derived from publicly available sources, with appropriate handling of sensitive domains such as healthcare. We ensure responsible data curation and do not redistribute any content that may violate licensing terms or privacy safeguards. Further details on ethical practices, licensing, and data access policies are provided in Appendix G.

## 3 IMM-TSF: A Benchmark Library for Irregular Multimodal Multivariate Time Series Forecasting

### 3.1 Problem Formulation

In standard irregular unimodal time series forecasting (TSF), we aim to predict future values of a numerical sequence using its past, irregularly sampled observations. Let $X = \{[(t_i^{(n)}, x_i^{(n)})]_{i=1}^{P_n}\}_{n=1}^N$, where there are $N$ variables and the $n$-th variable contains $P_n$ past observations. Each time series observation $x_i^{(n)} \in \mathbb{R}$ occurs at time $t_i^{(n)} \in [T_{\text{start}}, T_{\text{cut}}]$, where $T_{\text{start}}$ denotes the start time and $T_{\text{cut}}$ denotes the cut-off time separating past from future data. We define a set of future query times as $Q = \{[q_k^{(n)}]_{k=1}^{F_n}\}_{n=1}^N$, where each $q_k^{(n)} \in (T_{\text{cut}}, T_{\text{end}}]$ and $F_n$ denotes the number of future queries for variable $n$. The forecasting objective is to learn a function $f_\theta : (X, Q) \mapsto Y$, where the predictions are given by $Y = \{[\hat{y}_k^{(n)}]_{k=1}^{F_n}\}_{n=1}^N$, with each $\hat{y}_k^{(n)}$ approximating the true time series value at time $q_k^{(n)}$.

To incorporate unstructured textual information, we extend the irregular TSF setup to a multimodal setting. In addition to $X$, we observe a sequence of time-stamped textual data $S = \{(\tau_j, s_j)\}_{j=1}^{L_S}$, where $s_j$ is a text input observed at time $\tau_j \in [T_{\text{start}}, T_{\text{cut}}]$, and $L_S$ denotes the number of text observations. Here, $T_{\text{start}}$ is the shared start time for both the time series and the text modality, and no alignment is assumed between the time series timestamps $\{t_i^{(n)}\}$ and the text timestamps $\{\tau_j\}$. The multimodal forecasting objective is to learn a function $g_\theta : (X, S, Q) \mapsto Y$ that integrates the temporal dynamics of the irregular time series and contextual signals from the text to produce accurate predictions $Y = \{[\hat{y}_k^{(n)}]_{k=1}^{F_n}\}_{n=1}^N$ at the query times $Q \subset (T_{\text{cut}}, T_{\text{end}}]$. An overview of this multimodal irregular forecasting setup is illustrated in Figure 3.

### 3.2 Multimodal TSF Library

To support research on forecasting under realistic multimodal irregularity, we introduce IMM-TSF, a modular benchmark library designed for plug-and-play experimentation. An overview of the library architecture is shown in Figure 4. IMM-TSF supports flexible composition of forecasting pipelines by exposing unified interfaces for numerical encoders, textual encoders, and fusion strategies. Each

component is designed to be self-contained and easily swappable, allowing researchers to explore different architectural choices without needing to modify the rest of the pipeline.

**Irregular Unimodal TSF Model**   IMM-TSF supports both irregular time series models and regular forecasting architectures adapted to irregular data. To adapt regular models without imputing away irregularity, we apply a canonical pre-alignment strategy following t-PatchGNN [78]. Pre-alignment converts variable-length, unevenly spaced inputs into fixed-length tensors while preserving timestamps and missingness masks, allowing models to learn directly from irregular observation patterns. Specifically, observed values, binary masks, and normalized timestamps are aligned to a global temporal grid, and future query times are appended as unobserved points to guide inference. This approach offers a transparent, assumption-free way to reuse existing forecasting architectures under irregular sampling, unlike Gaussian Process–based methods that perform full imputation before modeling. Further algorithmic details and an illustrative example are provided in Appendix H.

**Text Encoder**   Textual inputs are processed using pretrained language models, which are frozen during training to prevent overfitting. IMM-TSF supports a variety of open-source LLMs and compact encoders, including GPT-2 [53], BERT [16], Llama-3.1-8B [19], and DeepSeek-7B [5]. The resulting embeddings form the asynchronous textual input stream for downstream fusion.

**Timestamp-to-Text Fusion (TTF) Module**   In real-world settings, textual data typically arrives asynchronously and is not temporally aligned with the timestamps of numerical time series. The Timestamp-to-Text Fusion module addresses this by using the timestamps of past textual data to produce temporally aware representations for each forecast query. IMM-TSF provides two variants. (1) *Recency-Weighted Averaging (RecAvg)* performs a Gaussian-weighted aggregation of past text embeddings, where weights decrease with the squared time difference between the text and forecast timestamps. This allows the model to emphasize temporally proximate text while remaining simple and efficient. (2) *Time2Vec-Augmented Cross-Attention (T2V-XAttn)* enriches each past text embedding with a Time2Vec [29] encoding of its timestamp, then uses cross-attention with a learned forecast query to selectively attend to temporally and semantically relevant past text.

**Multimodality Fusion (MMF) Module**   The Multimodality Fusion module combines text-derived representations with numerical features before prediction, allowing textual context to directly influence the final forecast. IMM-TSF includes two variants. (1) *GRU-Gated Residual Addition (GR-Add)* encodes a residual correction from the concatenated text and numerical features using a GRU, and blends it into the forecast via a learned gating mechanism that adapts the contribution of text dynamically. (2) *Cross-Attention Addition (XAttn-Add)* applies multi-head attention between numerical queries and text representations, and adds a scaled residual update to the forecast using a convex combination controlled by a fixed mixing weight. Further architectural details for both TTF and MMF modules are provided in Appendix I.

## 4   TSF Experiments on Irregular Multimodal Multivariate Time Series

### 4.1   Experimental Setup

**Datasets and Forecasting Setup.**   We evaluate all models on the nine TIME-IMM datasets, each corresponding to a distinct cause of real-world time series irregularity. Each dataset contains multiple entities (e.g., patients, devices, repositories), and we define an entity-specific forecasting task, where both the input and target windows are sampled from the same entity without crossing entity boundaries. We apply a sliding window approach, dividing each window into a past (context) segment and a future (query) segment, enabling models to learn from historical observations and predict future values. Data is split chronologically into 60% training, 20% validation, and 20% test. Window sizes for both context and query segments are dataset-specific, reflecting native timestamp distributions and sampling patterns. Further configuration details are provided in Appendix J.

**Baselines.**   We compare a broad range of models categorized into three groups: (1) *Regular Time Series Forecasting Models*, including Informer [81], DLinear [76], PatchTST [47], TimesNet [69], and TimeMixer [65]; (2) *Large Time Series Models*, including the LLM-based TimeLLM [26] and the foundation model TTM [20]; and (3) *Irregular Time Series Forecasting Models*, including CRU [57],

Latent-ODE [56], Neural Flow [6], and t-PatchGNN [78]. We evaluate each model in both unimodal and multimodal settings, using mean squared error (MSE) as our primary performance metric.

**Training and Optimization.**    All models are trained using the Adam optimizer with a learning rate of 1e-3 and a batch size of 8. Training proceeds until early stopping based on validation loss. For model-specific hyperparameters such as the number of layers or hidden dimensions, we use the default settings provided by the original implementations. In multimodal settings, we use the best-performing combination of text encoder, Timestamp-to-Text Fusion (TTF) module, and Multimodality Fusion (MMF) module selected via validation performance. Full details for each baseline and multimodal configuration are provided in Appendix K.

To further examine temporal robustness, we conduct rolling-origin cross-validation to evaluate generalization across different time windows. Models trained on TIME-IMM show stable forecasting performance under temporal drift, demonstrating strong generalization across time. Detailed results are provided in Appendix P.

## 4.2    Results and Analysis

**Effectiveness of Multimodality.**    Across all baselines, we observe consistent performance gains when incorporating textual signals alongside irregular time series data. Figure 5 shows that multimodal variants outperform unimodal counterparts in nearly every case. While prior work like Time-MMD demonstrated the value of text in regular time series forecasting, our results extend this insight to irregular settings. On average, the best multimodal configurations reduce MSE by 6.71%, with gains reaching up to 38.38% in datasets with informative text—confirming that textual context offers complementary cues under real-world irregularities. Detailed results are provided in Appendix L. Among irregular models, t-PatchGNN achieves the strongest results—consistent with its status as the state-of-the-art in irregular unimodal forecasting—and further validates the quality of our carefully curated irregular benchmark. We also find that models suited to small-scale data—such as the few-shot capable TimeLLM and TTM, and the lightweight DLinear—remain

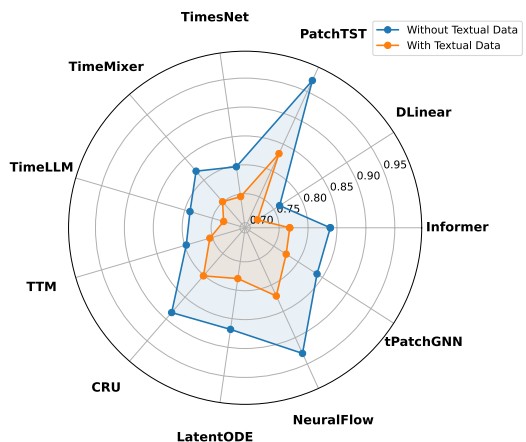

Figure 5: Radar chart comparing forecasting performance across all baseline models. Shaded regions highlight the relative improvement from multimodal over unimodal variants.

competitive in the multimodal setting. This may be due to the smaller size of multimodal datasets compared to unimodal ones, where models that are both simple and possess strong few-shot capabilities are easier to train effectively.

**Impact of Irregularity Patterns.**    Forecasting difficulty in the unimodal setting varies substantially across different subtypes of time series irregularity. Figure 6a plots per-dataset forecasting errors across baselines, grouped by the irregularity subtypes defined in our taxonomy. Datasets characterized by Unplanned Missing Data (ILINet), Resource-Aware Collection (ClusterTrace), or Event-Based Logging (GDELT) are notably more difficult, as their observation patterns are erratic or driven by latent semantic triggers that are inherently harder to model. In contrast, datasets such as RepoHealth (Adaptive Sampling), FNSPID (Operational Window Sampling), and EPA-Air (Multi-Source Asynchrony) tend to be easier to model due to more predictable or recoverable observation structures. For instance, in the FNSPID dataset, which involves stock price forecasting, the irregularity stems from known business-hour trading windows—an expected pattern that simplifies temporal reasoning.

When extending to the multimodal setting, we find that performance gains from incorporating text are not determined solely by the type of irregularity. Instead, improvements vary by dataset and are closely tied to the predictive value of the textual modality. For instance, in FNSPID, where numerical forecasting is already strong due to the regularity of stock market hours, textual news adds limited

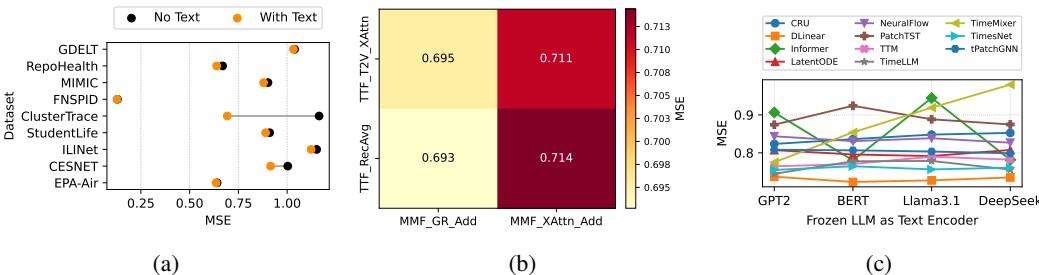

Figure 6: Multimodal forecasting analysis: (a) gains across datasets; (b) fusion strategies; (c) frozen LLM backbones.

additional benefit. Conversely, ClusterTrace sees substantial gains from textual integration because its workload annotations describe the currently running processes, which are directly relevant for forecasting future resource usage. These findings highlight that the value of textual data in multimodal irregular forecasting depends more on the semantic alignment and predictive strength of the text than on the irregularity pattern itself.

**Ablation of Fusion Modules.**   To evaluate the design choices within our fusion framework, we conduct ablations over the Timestamp-to-Text Fusion (TTF) and Multimodality Fusion (MMF) modules, as summarized in Figure 6b. For TTF, we observe no notable performance difference between the Time2Vec-augmented cross-attention (T2V-XAttn) and the simpler recency-based averaging (RecAvg), suggesting that both strategies for handling irregular timestamps in textual data are comparably effective. For MMF, GRU-gated residual addition (GR-Add) consistently outperforms cross-attention addition (XAttn-Add), suggesting that learnable gating provides a more effective means of integrating text-derived context into time series representations. By selectively controlling the flow of information, the gating mechanism in GR-Add helps suppress noisy or less relevant textual signals, leading to more stable and robust fusion. A detailed comparison of computational cost and efficiency across different TTF and MMF module configurations is provided in Appendix N.1.

**Role of the Text Encoder.**   We evaluate the impact of different LLM backbones on forecasting performance in the irregular multimodal TSF setting by varying the text encoder within IMM-TSF. Specifically, we test GPT-2, BERT, Llama 3.1 (8B), and DeepSeek (7B), and assess their performance across representative datasets. As shown in Figure 6c, we observe no significant performance differences across these models, suggesting that the choice of LLM backbone does not strongly influence results under irregular conditions. This lack of sensitivity may reflect the nature of the irregular TSF task: because forecasting hinges more on temporal alignment and contextual anchoring than on deep semantic reasoning, increasing LLM capacity offers no clear benefit. Additionally, while our TTF and MMF modules enable integration of textual signals into the forecasting pipeline, the overall scale of our datasets remains relatively small compared to the large-scale data used to pretrain LLMs, which may limit the advantages of larger models in this setting. Further experiments, including a lightweight encoder trained from scratch (Doc2Vec) and evaluations with very large LLMs (for example, Llama-70B), are presented in Appendix M, while a detailed analysis of text encoder efficiency is provided in Appendix N.2.

## 5   Future Work

**Forecasting Without Known Query Timestamps.**   Following prior work such as t-PatchGNN [78], we assume that forecast query times are given, simplifying the task to predicting values at predefined future timestamps. A natural extension is to consider more open-ended settings where the model must first infer when important events or state changes are likely to occur, and then predict their values. This setup—akin to timestamp prediction or event forecasting—introduces additional uncertainty and demands joint modeling of temporal dynamics and event saliency.

**Beyond Forecasting Tasks.**   While IMM-TSF currently focuses on forecasting as the primary task, real-world irregular multimodal time series often require broader capabilities such as anomaly

detection, classification, or retrieval. Adapting our framework to support these tasks would enable more comprehensive evaluation of modeling techniques and open new directions for multimodal irregular time series learning under operational constraints.

**Beyond Textual Modalities.** Our current benchmark focuses on unstructured text as the secondary modality, but many real-world applications involve other asynchronous inputs such as images, audio, or tabular data. Extending irregular time series datasets to support multimodal integration beyond text remains an important direction for building truly general-purpose time series models. As a proof of concept, we demonstrate in Appendix O that IMM-TSF can be readily extended to incorporate image-based modalities without architectural changes, highlighting its flexibility as a foundation for future multimodal extensions.

**Compositional Irregularities.** Each dataset in TIME-IMM currently represents a single, well-defined irregularity pattern to ensure interpretability and controlled benchmarking. A natural next step is to support datasets that combine multiple irregularity sources, reflecting the complex conditions found in real-world systems. We are developing APIs for compositional irregularity generation, allowing users to specify and mix atomic patterns (for example, human scheduling with missing data) to create hybrid datasets with richer temporal dynamics and more challenging forecasting scenarios.

## 6  Conclusion

We introduce TIME-IMM, the first benchmark explicitly designed around cause-driven irregularities in multimodal multivariate time series. Our taxonomy covers three distinct categories of irregularity—trigger-based, constraint-based, and artifact-based—captured across nine real-world datasets. To complement this, we present IMM-TSF, a plug-and-play library for forecasting on irregular multimodal time series, with modular support for textual encoders and fusion strategies. Empirical results show that integrating textual information improves forecasting accuracy across all irregularity types, underscoring the need for robust multimodal modeling under realistic conditions. TIME-IMM and IMM-TSF are positioned to serve as strong foundations for advancing time series research toward more general and practically relevant solutions.

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

# A Related Dataset Work

Recent progress in time series analysis has been supported by a growing number of publicly available benchmarks [67, 43, 52, 59]. However, most existing datasets make simplified assumptions that limit their realism and utility for modeling complex real-world phenomena. In particular, prior time series datasets often fall short along several key dimensions that TIME-IMM is explicitly designed to address.

**Lack of Irregularity.** Many standard time series datasets assume regularly sampled data where all features are observed at fixed intervals [81, 55, 49, 21]. Even in multimodal contexts, existing benchmarks often maintain regular sampling, smoothing over the irregular behaviors common in real-world systems [41, 38]. This assumption neglects the complexities found in domains like healthcare, environmental sensing, and finance, where sampling can be triggered by events, human decisions, or operational constraints [77, 73, 15].

**Lack of Asynchronous Textual Modality.** Most existing time series datasets include only numerical observations and lack any unstructured textual modality [14, 44]. While some benchmarks incorporate alternative modalities such as images, audio, or video [1, 3, 25, 71], this work focuses specifically on unstructured textual data as the complementary modality to time series. Yet in many domains, textual signals—such as clinical notes, system logs, or policy descriptions—are crucial for interpretability and context-aware forecasting [8, 17, 7]. Even in datasets that include both time series and text [41, 79], the modalities are often artificially aligned by enforcing shared timestamps, disregarding the fact that different modalities in real-world settings typically follow inherently asynchronous temporal patterns. Our work explicitly supports asynchronous timestamps across modalities, preserving their natural temporal structure for more realistic modeling.

**Lack of Reasoning About Irregularity Causes.** While some prior datasets contain irregularly sampled time series [78, 60, 62, 45], they do not provide any structured explanation or annotation of why the irregularity occurs. Irregular sampling is typically treated as a nuisance to be smoothed or interpolated away, rather than a meaningful signal that could inform model design or interpretation [30, 64, 74]. This lack of causal reasoning limits model robustness and interpretability, especially in high-stakes domains such as healthcare, environmental monitoring, or fault diagnostics [4, 51]. In contrast, our work introduces a structured taxonomy of irregularity, categorizing it into trigger-based, constraint-based, and artifact-based causes to support more interpretable and realistic modeling.

# B Limitations

While our proposed dataset and benchmark provide a valuable foundation for studying irregular multimodal time series forecasting, several limitations remain.

First, the current version of our dataset focuses exclusively on English-language textual data. This choice simplifies preprocessing and alignment but limits the applicability of our benchmark in multilingual or cross-lingual settings, which are common in global-scale systems such as public health surveillance or international finance [24, 2, 80].

Second, our benchmark is currently limited to the task of time series forecasting. We chose this task in part because it does not require additional label collection—making it easier to construct realistic datasets from naturally occurring logs or signals [9, 11]. Moreover, forecasting serves as a critical upstream task in time series analysis: models that accurately capture future dynamics can support a range of downstream applications including anomaly detection, classification, and decision support [39, 75]. Nonetheless, expanding the benchmark to cover these broader tasks would enable more comprehensive evaluation of modeling techniques.

Third, our focus on textual modality reflects the increasing utility of natural language as a rich, unstructured source of auxiliary information. This design decision is motivated by the recent success of large language models (LLMs), which have demonstrated strong capabilities in extracting and contextualizing semantic information from text. However, we do not currently support other modalities such as images, audio, or tabular records—despite their relevance in domains like remote sensing, clinical medicine, or human activity recognition [33, 34, 23].

Fourth, we use pretrained LLMs as frozen text encoders and do not explore task-specific fine-tuning. This design simplifies benchmarking and reduces computational cost, but it may underutilize the

full potential of the language models, especially in domains where domain-specific adaptation could improve alignment with time series signals [42, 66]. To isolate the value of pretrained representations, we conducted additional experiments using Doc2Vec [35] as a train-from-scratch alternative. These results, presented in Appendix M.1, confirm that frozen LLMs consistently outperform Doc2Vec, reinforcing their utility even without fine-tuning.

Finally, while we introduce modular mechanisms for timestamp-aware fusion of text and time series, we do not explore more advanced strategies such as retrieval-augmented generation (RAG) [72, 48], memory modules [37, 22], or reasoning [13, 70], which could further enhance model interpretability and performance.

Despite these limitations, our work offers a structured and extensible platform for modeling real-world time series data under conditions of irregularity and multimodality, and we hope it encourages future research into more general, robust, and semantically grounded forecasting systems.

## C Formalization of Irregularity Metrics

We introduce three metrics to quantitatively characterize the irregularity of time series data across entities and modalities. These metrics are designed to be interpretable, normalized where appropriate, and sensitive to structural patterns in missingness and temporal spacing.

### C.1 Feature Observability Entropy

This metric quantifies how evenly features are observed across the dataset. Let $N$ be the number of features, and $f_i$ the number of non-missing observations for feature $i$. Define the proportion of observations for each feature as:

$$p_i = \frac{f_i}{\sum_{j=1}^{N} f_j} \tag{1}$$

The normalized entropy is computed as:

$$\hat{H}_{\text{feat}} = -\frac{1}{\log N} \sum_{i=1}^{N} p_i \log(p_i + \epsilon) \tag{2}$$

where $\epsilon$ is a small constant added for numerical stability.

**Interpretation:** A higher value of $\hat{H}_{\text{feat}}$ (closer to 1) indicates that observations are evenly distributed across all features. A lower value (closer to 0) suggests that only a subset of features dominate the observation space, revealing imbalance in feature observability.

### C.2 Temporal Observability Entropy

This metric measures the dispersion of observations over time by dividing the full time span into $K$ equal-width bins and evaluating the entropy of observation counts within each bin. Let the total time range be divided into $K$ bins of equal width, and let $c_k$ be the number of observations falling into bin $k$. The normalized count in each bin is:

$$p_k = \frac{c_k}{\sum_{j=1}^{K} c_j} \tag{3}$$

The normalized temporal entropy is computed as:

$$\hat{H}_{\text{temp}} = -\frac{1}{\log K} \sum_{k=1}^{K} p_k \log(p_k + \epsilon) \tag{4}$$

where $\epsilon$ is a small constant added for numerical stability. We fix $K = 10$ across all datasets to provide a stable and consistent measure of temporal dispersion.

**Interpretation:** A higher value of $\hat{H}_{\text{temp}}$ (closer to 1) indicates that observations are evenly distributed across time, suggesting regular sampling behavior. A lower value (closer to 0) reflects temporal clustering or bursty updates.

### C.3 Mean Inter-Observation Interval (IOI)

This metric quantifies the average time gap between consecutive observations.

Let $\Delta t_i = t_i - t_{i-1}$ denote the interval between successive timestamps across all entities, and let $M$ be the total number of such intervals. The mean IOI is:

$$\text{Mean IOI} = \frac{1}{M} \sum_{i=1}^{M} \frac{\Delta t_i}{s} \tag{5}$$

where $s$ is the number of seconds per time unit (e.g., 86,400 for days).

**Interpretation:** Smaller values indicate denser data (frequent sampling), while larger values suggest sparser time series. This metric is unnormalized and reported in interpretable units (e.g., hours or days).

## D  Dataset Details and Data Sources

We organize Time-IMM's nine datasets according to our irregularity taxonomy. For each, we describe the data, point to the source, and summarize how numerical and textual streams were collected.

### D.1  Event-Based Logging (Trigger-Based)

**Description**  Observations occur only when external events trigger data collection. This pattern reflects systems that respond to real-world happenings such as political crises, protests, or policy changes.

**Source**  GDELT Event Database 2.0 (2021–2024) [36]. Raw data was downloaded from: `http://data.gdeltproject.org/gdeltv2/masterfilelist.txt`

**Entity**  Each entity corresponds to a unique (country, event type) pair. We select 4 countries—`USA`, `GBR`, `AUS`, and `CAN`—and 2 event types: `AGR` (Agriculture) and `LAB` (Labor Union). This results in 8 distinct entities in total.

**Collection**

- **Numerical Time Series:** We use GDELT's event dataset from 2021/01/01 to 2024/12/31 and extract five key quantitative features per event: `GoldsteinScale`, `NumMentions`, `NumSources`, `NumArticles`, and `AvgTone`. Events are grouped by (country, type) to create entity-specific time series.
- **Textual Data:** For each event, we download the associated news article using the source URL provided in GDELT. These articles are filtered for relevance and summarized into five-sentence narratives using GPT-4.1 Nano, following prompt instructions designed to ensure factuality, brevity, and contextual clarity.

### D.2  Adaptive or Reactive Sampling (Trigger-Based)

**Description**  Sampling frequency is adjusted dynamically based on system activity levels. In this case, repository activity determines when state is logged—shorter intervals during development bursts and longer intervals during quiet periods.

**Source**   We use GitHub repository data from the GitHub Archive via Google BigQuery.

**Entity**   Each entity corresponds to a single GitHub repository. This results in four total entities in the dataset:

1. `facebook/react`
2. `huggingface/transformers`
3. `kubernetes/kubernetes`
4. `pytorch/pytorch`

**Collection**

- **Numerical Time Series:** We compute a 10-dimensional numerical time series from GitHub metadata, including issue activity, pull request lifecycle statistics, backlog growth, and age metrics. The data is sampled on an adaptive schedule determined by commit frequency: 1 day during high activity, 7 days for baseline, and 14 days during low activity. Each timestamp reflects a state summary at the chosen interval.

- **Textual Data:** For each sampled interval, we gather all issue and pull request comments made since the previous timestamp. We limit the number of comments to 5 per issue to control input length. These comments are aggregated by repository and date, then summarized using GPT-4.1 Nano with a prompt designed to extract key concerns, proposals, and unresolved questions. The generated summaries consist of up to five sentences and serve as the unstructured modality aligned with each adaptive sample.

### D.3   Human-Initiated Observation (Event-Based)

**Description**   Data is recorded when humans initiate measurement or documentation, such as clinicians logging vitals based on clinical necessity. This results in inherently irregular sampling tied to human judgment and workflow.

**Source**   MIMIC-IV clinical database and MIMIC-Notes [27, 28]. Available at: `https://physionet.org/content/mimiciv/3.1/`

**Entity**   Each entity is a single ICU patient who satisfies the following inclusion criteria:

- Patients included in the **Metavision** system.
- Exactly **one hospital admission** (to avoid multi-stay confounds).
- **Length of stay** exceeds **50 days**, ensuring sufficiently long time-series.
- **Age** at admission is $\geq 15$ years.
- Has at least one chart-events record and appears in the **MIMIC-IV Notes** table.

From the remaining population we draw a simple random sample of $n = 20$ patients to capture diverse clinical courses and recording patterns. The choice of criteria and the preprocessing code mostly follow the logic from the GRU-ODE-Bayes preprocessing pipeline: `https://github.com/edebrouwer/gru_ode_bayes/tree/master/data_preproc/MIMIC`

**Collection**

- **Numerical Time-Series**
  For each selected patient we extract 105 variables spanning: IV infusion and boluses, laboratory chemistry and hematology, blood products and fluids, outputs such as urine and stool. Measurements are documented only when clinically indicated, yielding patient-specific, irregular sampling.

- **Textual Data**
  All free-text notes (e.g. progress, nursing, radiology) linked to the same patients are retained in raw form together with their original timestamps. These notes provide asynchronous narrative context that can precede, coincide with, or follow numerical measurements.

This protocol produces a multimodal dataset—irregular physiological signals paired with temporally aligned clinical narratives—tailored for longitudinal modelling tasks.

## D.4   Operational Window Sampling (Constraint-Based)

**Description**   This form of irregularity occurs when observations are limited to predefined operational hours, such as trading windows on financial markets. No data is collected during weekends, holidays, or outside business hours, resulting in structured temporal gaps.

**Source**   FNSPID: Financial News & Stock Prices [18].

**Entity**   Each entity corresponds to an individual publicly traded company. We first filter out companies that lack recorded time stamps or related articles, and then randomly sample 10 different companies.

**Collection**

- **Numerical Time Series:** We download daily closing prices and related trading statistics for each selected company from `https://huggingface.co/datasets/Zihan1004/FNSPID`. Only trading days are included, with no entries for weekends or market holidays, leading to regularly spaced but discontinuous sampling.

- **Textual Data:** For each trading day, we collect news headlines and article excerpts related to the corresponding company from `https://huggingface.co/datasets/Zihan1004/FNSPID`. Articles are time-stamped and filtered for relevance. We then use GPT-4.1 Nano to summarize the article into concise 5-sentence narratives, highlighting market developments and company-specific events.

## D.5   Resource-Aware Collection (Constraint-Based)

**Description**   This irregularity arises when data is recorded only while compute resources are active. In this setting, telemetry is missing not due to failure or noise, but because the monitoring system is deliberately inactive during idle periods.

**Source**   Alibaba GPU Cluster Trace v2020 [68]. Main repository: `https://github.com/alibaba/clusterdata/tree/master?tab=readme-ov-file`

**Entity**   Each entity corresponds to a distinct physical GPU machine in the Alibaba cluster. We select 3 machines from the dataset, identified by their anonymized `worker_name`, to represent heterogeneous task schedules and resource usage patterns.

**Collection**

- **Numerical Time Series:** We process eight weeks of Alibaba GPU cluster logs from July–August 2020. For each selected machine, we generate a time series where measurements are sampled only during active periods (i.e., when at least one container is running). At each event timestamp, we log:
  - The number of concurrent instances, tasks, and jobs
  - Eight sensor metrics aggregated across all currently active containers

  The result is an 11-dimensional time series with naturally irregular intervals, reflecting operational constraints and runtime dynamics.

- **Textual Data:**   For every container start event, we extract the corresponding `job_name`, `task_name`, `gpu_type_spec`, and `workload` description from the `pai_group_tag_table.csv`. These fields are treated as an asynchronous textual context. We aggregate all textual entries within a 6-hour window and use GPT-4.1 Nano to summarize them into concise narratives of no more than 5 sentences, capturing key workload patterns and system activity during that period.

### D.6 Human Scheduling / Availability (Constraint-Based)

**Description**   This form of irregularity occurs when data collection is driven by human activity patterns or availability—such as sensors only logging when users are awake or interacting with their devices.

**Source**   StudentLife study (College Experience Dataset) [46].

**Entity**   Each entity corresponds to an individual participant in the study. We first filter out students with less than 1000 records, then randomly select 20 students from the full cohort to represent a diverse range of behavioral patterns and schedules.

**Collection**

- **Numerical Time Series:** We use smartphone sensor logs—including activity durations on e.g., bike, foot, sleep—recorded at naturally irregular intervals driven by device usage, battery state, and user activity.
- **Textual Data:** We extract self-reported survey responses (e.g., anxiety, depression, stress level), each time-stamped by the participant's device. These serve as asynchronously generated text signals that reflect personal context, mood, or behavior at different points in time.

### D.7 Unplanned Missing Data/Gaps (Artifact-Based)

**Description**   This irregularity occurs when expected data points are unintentionally missing due to system failures, transmission errors, or reporting issues. Unlike intentional missingness from adaptive sampling or human decisions, these gaps arise without design and are usually treated as random (MCAR or MAR) and handled by imputation.

**Source**   ILINet influenza surveillance (CDC) [41].

**Entity**   This dataset contains a single entity: the United States as a national-level reporting unit. All observations are aggregated at the country level.

**Collection**

- **Numerical Time Series:** We download weekly counts of influenza-like illness (ILI) cases as reported by the CDC through ILINet. When reporting is incomplete or unavailable for a given week, we preserve these as missing values (NaNs) to reflect true observational gaps.
- **Textual Data:** We include weekly CDC public health summaries and provider notes curated from the report and search result collection process introduced by Time-MMD [41]. These documents are treated as asynchronous textual observations and provide contextual insight into public health events, seasonal flu trends, and potential reporting disruptions.

### D.8 Scheduling Jitter / Delay (Artifact-Based)

**Description**   This irregularity arises when data points are recorded at uneven intervals due to system delays, congestion, or throttling. Instead of arriving on a regular schedule, observations appear with jitter caused by internal scheduling constraints or resource contention.

**Source**   CESNET network flow data [32].

**Entity**   Each entity corresponds to a distinct networked device in the infrastructure. We include 11 entities in total, spanning various device types such as IP endpoints, firewalls, routers, servers, and switches.

**Collection**

- **Numerical Time Series:** We ingest flow-level metrics such as byte and packet counts using real-world arrival timestamps. These metrics naturally exhibit jitter due to internal logging delays, variable sampling rates, and system load effects.
- **Textual Data:** We collect network device logs that include maintenance messages, traffic control alerts, and system warnings. Each log entry is treated as an independent, asynchronously occurring textual observation, preserving its original timestamp without forcing alignment to numeric data.

### D.9 Multi-Source Asynchrony (Artifact-Based)

**Description**    This type of irregularity arises when multiple data streams—such as sensors—operate with different clocks, sampling rates, or synchronization policies. These asynchronous signals pose challenges for fusion and temporal alignment, especially when modalities are collected independently.

**Source**    EPA Outdoor Air Quality Sensors [61]. Numerical time series data was downloaded from: `https://aqs.epa.gov/aqsweb/airdata/download_files.html`

Textual news articles were retrieved from Common Crawl News, using monthly archives available at: `https://data.commoncrawl.org/crawl-data/CC-NEWS/`

**Entity**    Each entity corresponds to a U.S. county monitored by the EPA sensor network. We select 8 counties for inclusion in this dataset:

- Los Angeles, CA
- Maricopa (Phoenix), AZ
- Philadelphia, PA
- Bexar (San Antonio), TX
- Dallas, TX
- Richmond, VA
- Hillsborough (Tampa), FL
- Denver, CO

**Collection**

- **Numerical Time Series:** We extract four environmental variables from the EPA's air monitoring dataset: `AQI`, `Ozone`, `PM2.5`, and `Temperature`. Each feature is recorded at its native sensor-specific frequency, resulting in asynchronous observations across features and counties.
- **Textual Data:** To provide contextual background for environmental conditions, we collect news articles from Common Crawl spanning January to October 2024. Articles are filtered to include only those that mention both the county name and the keyword "weather." Relevant articles are summarized using GPT-4.1 Nano into concise 5-sentence narratives, highlighting key environmental developments or disruptions affecting the region.

## E    Prompt Designed for LLM Summarization

This appendix presents the prompt templates used with GPT-4.1 Nano to generate textual summaries for each dataset in Time-IMM. Each dataset employs a single-step, prompt-based pipeline that jointly performs filtering and summarization. During preprocessing, the model generates a concise five-sentence summary only when the input document is topically relevant; otherwise, it returns "NA," which automatically excludes the document. This unified approach eliminates the need for a separate relevance classifier and ensures consistent semantic filtering across domains. For instance, the prompts remove off-topic news in GDELT, retain technical discussions in RepoHealth, and focus on market-related news in FNSPID. A manual review of 250 sampled summaries showed 100% recall and 88.4% precision in identifying relevant documents.

### E.1 GDELT (Event-Based Logging)

```
PROMPTS = """
You are an expert in {domain}.
This task is part of building the GDELT 2.0 Event dataset.

Below is a news article:
{article}

Write a concise summary of the information related to {domain}, using no
    more than 5 sentences.
If the article is not relevant to {domain}, reply with exactly: NA
"""
```

### E.2 RepoHealth (Adaptive or Reactive Sampling)

```
PROMPTS = """
The following are comments from multiple GitHub issue threads in the
    repository '{repo_name}',
including all historical comments accumulated up to and including those
    made on a specific day.

{comments}

Write a concise summary of the main technical concerns, questions,
or suggestions raised across these discussions. Focus on key issues,
    proposed solutions,
and unresolved questions. Your summary should be no more than 5
    sentences.
"""
```

### E.3 FNSPID (Operational Window Sampling)

```
PROMPTS = """
You are an expert in financial markets.
This task is part of building the stock price prediction dataset.

Below is a news article:
{article}

Write a concise summary of the information related to financial markets,
    using no more than 5 sentences.
If the article is not relevant to financial markets, reply with exactly:
    NA
"""
```

### E.4 ClusterTrace (Resource-Aware Collection)

```
PROMPTS = """
You are an expert in large-scale distributed computing and cloud
    infrastructure.
This task is part of building the Alibaba Cluster Trace Program dataset.

Below is a collection of logs and system behavior summaries:
{article}

Write a concise summary of the system behavior and operational insights
    relevant to distributed computing, using no more than 5 sentences.
If the content is not relevant to the dataset, or the logs/summaries are
    not provided, reply with exactly: NA
"""
```

### E.5 EPA-Air (Multi-Source Asynchrony)

```
PROMPTS = """
You are an expert in weather.
This task is part of building the Air Quality Data Collected at Outdoor
    Monitors Across the US dataset.

Below is a news article:
{article}

Write a concise summary of the information related to weather, using no
    more than 5 sentences.
If the article is not relevant to weather, reply with exactly: NA
"""
```

## F  Showcase of Raw and Summarized Text

This appendix illustrates how raw textual inputs are preprocessed into concise, informative summaries using LLMs (GPT-4.1 Nano). Below is an example taken from the EPA-Air dataset, where the input news article contains broader event details, but the summarization process emphasizes weather conditions relevant to environmental sensing.

**EPA-Air Example**

**Raw News Excerpt:**

> Seven migrant smuggling suspects have been arrested and 11 migrants were hospitalized after a law enforcement sting Thursday near San Antonio, authorities said. ...
> Temperatures in San Antonio were in the high 90s Thursday afternoon and were expected to top 100, according to the National Weather Service.
> ...The incident came two days after President Joe Biden unveiled plans to enact immediate significant restrictions on migrants seeking asylum ...
> San Antonio was the site of the nation's deadliest human smuggling episode in June 2022 ...

**Summarized Version:**

On Thursday in San Antonio, temperatures reached the high 90s and were expected to exceed 100 degrees Fahrenheit. These hot conditions contributed to heat-related injuries among migrants found in a trailer during a law enforcement operation. The extreme heat poses a significant risk for dehydration, heat exhaustion, and other heat-related health issues. Past incidents, such as the deadly smuggling episode in June 2022, also occurred under similar sweltering conditions. The current weather highlights the dangers faced by individuals exposed to high temperatures in the area.

## G    Additional Ethical Considerations and Data Access

**Data Sources and Privacy.**    All datasets in TIME-IMM are derived from publicly available sources with appropriate access permissions. For sensitive domains such as healthcare (e.g., MIMIC-IV), we take additional precautions to ensure data privacy and regulatory compliance. MIMIC-IV is a restricted-access resource hosted on `https://physionet.org/content/mimiciv/3.1/`. To access the raw files, users must be credentialed, complete the CITI "Data or Specimens Only Research" training, and sign the official data use agreement on PhysioNet. In TIME-IMM, we do not redistribute raw data from MIMIC-IV. Instead, we provide preprocessing scripts and instructions to enable authorized users to reproduce our setup. We strictly follow access policies and do not feed MIMIC-IV clinical notes into closed-source LLMs (e.g., GPT-4.1 Nano). This ensures alignment with privacy safeguards and ethical standards for clinical data handling.

**Responsible Redistribution.**    To comply with upstream licenses and ethical research norms, we do not redistribute raw web-crawled text, GitHub comments, or clinical notes where direct sharing is restricted. Instead, we offer derived summaries, structured metadata, and links or instructions for dataset reconstruction from original public sources.

**Open Access and Licensing.**    TIME-IMM is released under a Creative Commons Attribution 4.0 (CC BY 4.0) license, which permits broad use, redistribution, and adaptation with appropriate citation. Scripts and benchmarks are openly available to ensure reproducibility, transparency, and extensibility by the research community.

**Hosting and Maintenance Plan.**    All datasets and code are hosted on publicly accessible and stable platforms. The dataset is maintained on Kaggle to facilitate broad usability, while the benchmark library is versioned and published through a persistent code repository. We commit to periodically reviewing contributions and issues reported by the community to ensure continued compatibility and correctness.

**Accessing.**    The TIME-IMM dataset can be accessed at: `https://www.kaggle.com/datasets/blacksnail789521/time-imm/data`

The accompanying benchmark code and processing tools are available at: `https://anonymous.4open.science/r/IMMTSF_NeurIPS2025`

**Ethical Usage Encouragement.**    We strongly encourage responsible usage of TIME-IMM and IMM-TSF in research contexts that prioritize fairness, transparency, and real-world safety. Applications involving downstream decision-making—especially in domains such as finance or healthcare—should incorporate appropriate validation and interpretability safeguards.

## H    Adapting Regular Time Series Models via Canonical Pre-Alignment

To adapt regular time series forecasting models to irregular data, we follow the canonical pre-alignment strategy introduced in t-PatchGNN [78], as illustrated in Figure 7. This method transforms irregularly sampled time series into a fixed-length, regularly spaced representation that can be consumed by standard forecasting architectures.

Let $X = \{[(t_i^{(n)}, x_i^{(n)})]_{i=1}^{P_n}\}_{n=1}^{N}$ denote the set of past irregular time series observations, where $N$ is the number of variables and $P_n$ is the number of past observations for variable $n$. Each observation

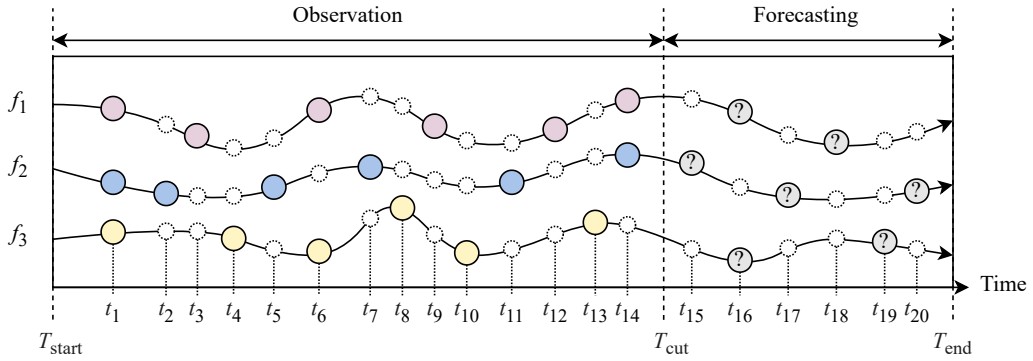

Figure 7: Overview of the canonical pre-alignment process.

$x_i^{(n)}$ occurs at time $t_i^{(n)} \in [T_{\text{start}}, T_{\text{cut}}]$. We also define a set of future query times $Q = \{[q_k^{(n)}]_{k=1}^{F_n}\}_{n=1}^{N}$, with each query $q_k^{(n)} \in (T_{\text{cut}}, T_{\text{end}}]$.

**Step 1: Determine Global Temporal Resolution.** We first scan the entire dataset to determine the global maximum number of past input timestamps and future query timestamps across all training windows. This defines the fixed-length resolution $L$ for both input and output time series, covering the combined past and forecast windows.

**Step 2: Temporal Grid Construction, Alignment, and Padding.** For each example, we create a unified, chronologically sorted list of timestamps $\{\tilde{t}_l\}_{l=1}^{L}$ that covers both the observation period and the forecast horizon. We construct a matrix $\tilde{X} \in \mathbb{R}^{L \times N}$, where each entry $\tilde{x}_l^{(n)}$ is the observed value of variable $n$ at time $\tilde{t}_l$, or zero if no observation exists. A binary mask matrix $M \in \{0, 1\}^{L \times N}$ is used to indicate whether a value was observed (1) or imputed (0). Both $\tilde{X}$ and $M$ are padded as necessary to reach the globally fixed length $L$, ensuring uniform tensor shapes across all examples.

**Step 3: Add Input Features: Timestamp and Mask.** To help the model reason about timing and missingness, we expand the feature dimension from $N$ to $2N + 1$. Each input row includes: (i) the observed values or zeros, (ii) the binary mask, and (iii) the normalized timestamp $\tilde{t}_l$. This preserves all temporal and structural information in a dense, aligned format.

**Step 4: Add Query Features.** The forecast queries $Q$ are appended to the end of the input sequence using the same $2N + 1$ feature schema. Each query timestamp $q_k^{(n)}$ is represented by a feature row that includes: (i) zero values for all variable entries (since ground truth is unknown at inference time), (ii) a binary mask set to 0 (indicating no observation—not a forecast target), and (iii) the normalized query timestamp. This mask entry should not be confused with the evaluation-time forecast mask used to compute prediction errors. Including query timestamps in the input allows the model to explicitly condition on the temporal locations of future targets.

**Summary.** This pre-alignment pipeline transforms irregular, variable-length time series into padded, fixed-length tensors that include time, observation, and mask features. By appending forecast queries and exposing the full temporal grid, regular forecasting models can be repurposed to handle irregular time series in a structured and query-aware manner.

# I  Formal Definitions of TTF and MMF Modules

The detailed architectures of the TTF and MMF modules are illustrated in Figures 8 and 9, respectively. We now present their formal definitions below.

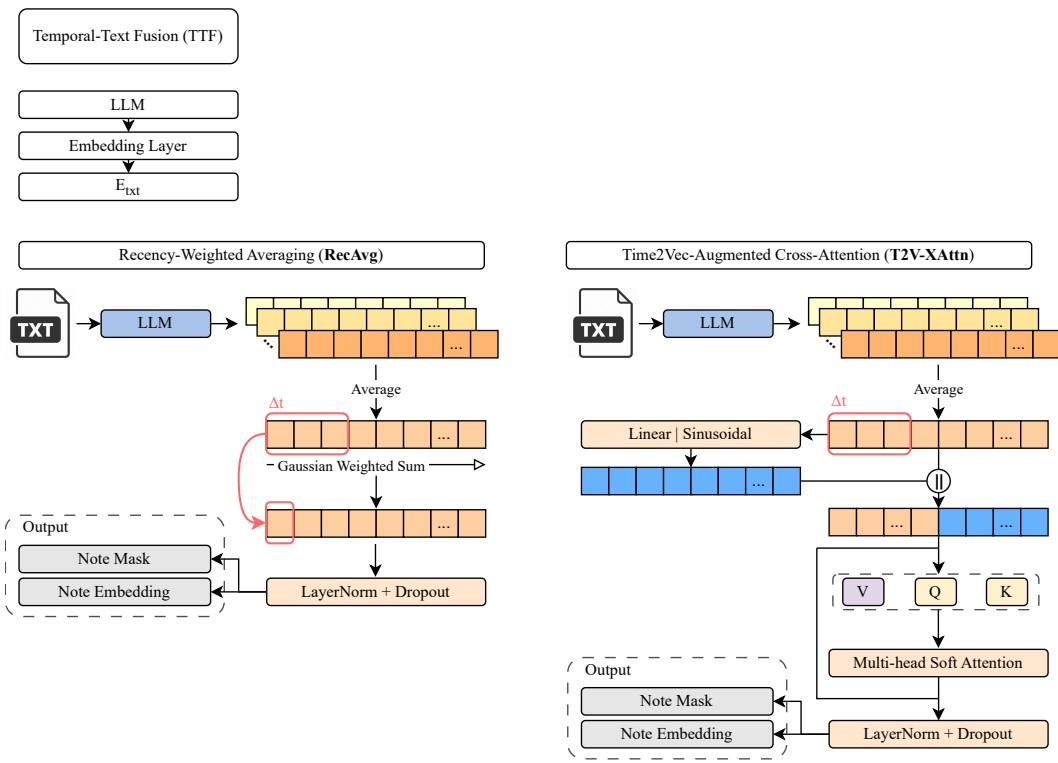

Figure 8: Detailed architecture of TTF modules.

## I.1 Timestamp-to-Text Fusion (TTF)

Let $S = \{(\tau_j, s_j)\}_{j=1}^{L_S}$ denote the sequence of past text observations, where $s_j$ is a text snippet observed at timestamp $\tau_j \in [T_{\text{start}}, T_{\text{cut}}]$. Let $v_j \in \mathbb{R}^d$ be the embedding of $s_j$ produced by a fixed language model. Let $Q = \{t_k\}_{k=1}^{T_f}$ denote the future forecast query times.

**Recency-Weighted Averaging (RecAvg).** This variant computes a time-weighted average of all previous embeddings using a Gaussian kernel centered at the forecast time:

$$\alpha_{jk} = \exp\left(-\left(\frac{t_k - \tau_j}{\sigma}\right)^2\right), \quad \text{for } \tau_j \le t_k \tag{6}$$

$$e_k = \frac{\sum_{j=1}^{L_S} \alpha_{jk} v_j}{\sum_{j=1}^{L_S} \alpha_{jk}} \in \mathbb{R}^d \tag{7}$$

**Time2Vec-Augmented Cross-Attention (T2V-XAttn).** Each embedding is augmented with a Time2Vec representation of its timestamp:

$$\phi(\tau_j) = [\omega_0 \tau_j + b_0, \ \sin(\omega_1 \tau_j + b_1), \dots, \sin(\omega_{d_\tau - 1} \tau_j + b_{d_\tau - 1})] \tag{8}$$

$$\tilde{v}_j = [\![v_j; \phi(\tau_j)]\!] \in \mathbb{R}^{d + d_\tau} \tag{9}$$

Let $W_q \in \mathbb{R}^{d_q \times 1}$ be a learnable forecast query vector. Then the aligned representation at time $t_k$ is computed by soft attention:

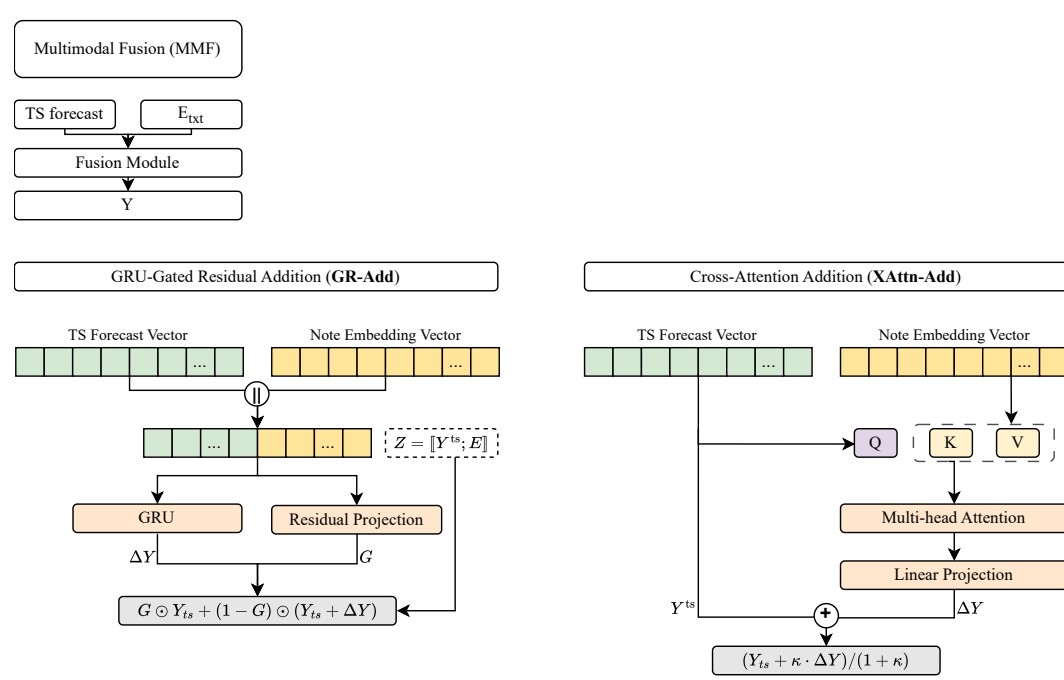

Figure 9: Detailed architecture of MMF modules.

$$a_{jk} = \frac{\exp\left((W_q^\top \tilde{v}_j)\right)}{\sum_{\tau_{j'} \leq t_k} \exp\left((W_q^\top \tilde{v}_{j'})\right)} \tag{10}$$

$$e_k = \sum_{\tau_j \leq t_k} a_{jk} \cdot \tilde{v}_j \tag{11}$$

## I.2  Multimodality Fusion (MMF)

Let $Y^{\text{ts}} \in \mathbb{R}^{T_f \times N}$ denote the numerical forecast sequence, and let $E = \{e_k\}_{k=1}^{T_f} \in \mathbb{R}^{T_f \times d}$ be the aligned text representations from TTF.

**GRU-Gated Residual Addition (GR-Add).**  Let $H \in \mathbb{R}^{T_f \times h}$ be the hidden state sequence from a GRU applied over the concatenated input:

$$z_k = [\![y_k^{\text{ts}}; e_k]\!] \in \mathbb{R}^{N+d} \tag{12}$$

$$H = \text{GRU}(\{z_k\}_{k=1}^{T_f}) \tag{13}$$

$$\Delta Y = W_\Delta H + b_\Delta \tag{14}$$

$$G = \sigma(W_g Z + b_g) \quad \text{where } Z = [\![Y^{\text{ts}}; E]\!] \tag{15}$$

$$Y^{\text{fused}} = G \odot Y^{\text{ts}} + (1 - G) \odot (Y^{\text{ts}} + \Delta Y) \tag{16}$$

where $W_\Delta \in \mathbb{R}^{N \times h}$, $W_g \in \mathbb{R}^{N \times (N+d)}$, and $\odot$ denotes element-wise multiplication.

**Cross-Attention Addition (XAttn-Add).**  Let $Q = Y^{\text{ts}} W_Q$, $K = E W_K$, and $V = E W_V$ with learnable projection matrices $W_Q, W_K, W_V$. The residual is derived via scaled dot-product attention:

Table 2: Dataset-specific forecasting parameters: context window (past input) and query horizon (forecast target).

| Dataset | Context Window | Query Horizon |
|---|---|---|
| GDELT | 14 days | 14 days |
| RepoHealth | 31 days | 31 days |
| MIMIC | 24 hours | 24 hours |
| FNSPID | 31 days | 31 days |
| ClusterTrace | 12 hours | 12 hours |
| StudentLife | 31 days | 31 days |
| ILINet | 4 weeks | 4 weeks |
| CESNET | 7 days | 7 days |
| EPA-Air | 7 days | 7 days |

$$A = \mathrm{softmax}\left(\frac{QK^{\top}}{\sqrt{d}}\right) \cdot V \tag{17}$$

$$\Delta Y = AW_{\mathrm{res}} + b_{\mathrm{res}} \tag{18}$$

$$Y^{\mathrm{fused}} = \frac{Y^{\mathrm{ts}} + \kappa \cdot \Delta Y}{1 + \kappa} \tag{19}$$

Here, $\kappa \in \mathbb{R}$ is a scalar hyperparameter, and all terms are defined over the entire sequence of query times.

## J  Dataset-Specific Forecasting Configuration

Each dataset in TIME-IMM is configured with a dataset-specific forecasting setup that reflects its native timestamp distribution and sampling behavior. For each entity, we apply a sliding window strategy with fixed-length context and query segments. The context window corresponds to past observations $X \in [T_{\mathrm{start}}, T_{\mathrm{cut}}]$, while the query horizon defines the future timestamps $Q \in (T_{\mathrm{cut}}, T_{\mathrm{end}}]$ at which the model is expected to make predictions. Durations are defined using dataset-native time units (e.g., hours, days, or weeks), ensuring semantic consistency across domains.

## K  Model-Specific and Fusion Module Configuration

### K.1  Model-Specific Configuration

Each baseline model is configured using recommended default hyperparameters from its original implementation. The following settings summarize key architecture and training parameters for each model class:

- **Informer** [81]: 2 encoder layers, 1 decoder layer, attention factor 3.
- **DLinear** [76]: Default configuration without modification.
- **PatchTST** [47]: 1 encoder layer, 1 decoder layer, 2 attention heads.
- **TimesNet** [69]: 2 encoder layers, 1 decoder layer, top-$k = 5$, $d_{\mathrm{model}} = 16$, $d_{\mathrm{ff}} = 32$.
- **TimeMixer** [65]: 2 encoder layers, $d_{\mathrm{model}} = 16$, $d_{\mathrm{ff}} = 32$, with 3 downsampling layers (window size 2, average pooling).
- **TimeLLM** [26]: GPT2 backbone with 6 layers, input/output token lengths 16 and 96, respectively; $d_{\mathrm{model}} = 32$, $d_{\mathrm{ff}} = 128$.

- **TTM** [20]: 3-level attention patching, 3 encoder and 2 decoder layers; $d_{\text{model}} = 1024$, decoder dimension 64.

- **CRU** [57]: Latent state and hidden units set to 32, with square/exponential activations and gravity gates enabled.

- **Latent-ODE** [56]: Recognition GRU with 32 hidden units, 1 RNN and 1 generator layer, ODE units set to 32.

- **Neural Flow** [6]: Latent dimension 20, 3-layer hidden network (dimension 32), GRU units 32, coupling flow with 2 layers, time network: TimeLinear.

- **tPatchGNN** [78]: Patch size 24, 1 Transformer layer, 1 GNN layer, 10-dimensional time and node embeddings, $d_{\text{hidden}} = 32$.

### K.2  Fusion Module Configuration

In multimodal experiments, we integrate textual context into the forecasting pipeline using specialized fusion modules. Each model uses the best-performing fusion configuration selected via validation performance.

**Timestamp-to-Text Fusion (TTF).**

- **RecAvg**: Recency-weighted averaging of past text embeddings using a Gaussian decay function based on timestamp proximity. The maximum weight is assigned when the text timestamp exactly matches the forecast time. We set the standard deviation parameter to $\sigma = 1.0$.

- **T2V-XAttn**: Each text embedding is augmented with a Time2Vec encoding of its timestamp and fused via a single-layer, single-head attention mechanism.

**Multimodality Fusion (MMF).**

- **GR-Add**: A residual correction is computed from the concatenated numerical forecast and text embedding using a GRU followed by an MLP. The output is blended with the base forecast through a learned gating function.

- **XAttn-Add**: Numerical forecasts attend to aligned text representations using a single-layer, single-head attention module. The result is scaled by a fixed convex mixing weight $\kappa = 0.5$ and added to the base forecast.

**Text Encoder.**  We utilize four kinds of pretrained large language models as text encoders—GPT-2, BERT-base, Llama 3.1 (8B), and DeepSeek (7B)—each used with its full transformer stack and no truncation of layers. BERT-base supports a maximum context length of 512 tokens, while GPT-2, Llama 3.1, and DeepSeek allow longer sequences but are forcibly truncated to 1024 tokens in our setup. To ensure consistent dimensionality, we project all encoder outputs to a shared 768-dimensional space using a learned linear transformation before fusion.

## L  Detailed Results on the Effectiveness of Multimodality

To quantify the contribution of textual signals under irregular sampling, we compare unimodal and multimodal model variants across all nine datasets in Time-IMM. Each result reflects the best-performing fusion configuration selected via validation on the corresponding dataset. Performance is measured using mean squared error (MSE), and improvements are reported as relative percentage change from the unimodal baseline.

Multimodal models consistently outperform their unimodal counterparts across all datasets, with especially large improvements observed in settings where text provides rich, temporally relevant context. The following tables summarize detailed results for each dataset:

- **GDELT (Event-Based Logging)** — Table 3
- **RepoHealth (Adaptive Sampling)** — Table 4

- **MIMIC (Human-Initiated Observation)** — Table 5
- **FNSPID (Operational Window Sampling)** — Table 6
- **ClusterTrace (Resource-Aware Collection)** — Table 7
- **StudentLife (Human Scheduling)** — Table 8
- **ILINet (Unplanned Missing Data)** — Table 9
- **CESNET (Scheduling Jitter)** — Table 10
- **EPA-Air (Multi-Source Asynchrony)** — Table 11

Table 3: Unimodal vs. multimodal forecasting results on the GDELT dataset.

| Model | Modal | MSE | MAE |
|---|---|---|---|
| Informer | Uni | 1.0448 | 0.6935 |
|  | Multi | 1.0295 | 0.6897 |
| DLinear | Uni | 1.0482 | 0.6943 |
|  | Multi | 1.0487 | 0.6945 |
| PatchTST | Uni | 1.0670 | 0.7051 |
|  | Multi | 1.0511 | 0.6944 |
| TimesNet | Uni | 1.0251 | 0.6846 |
|  | Multi | 1.0250 | 0.6916 |
| TimeMixer | Uni | 1.0471 | 0.6842 |
|  | Multi | 1.0333 | 0.6973 |
| TimeLLM | Uni | 1.0411 | 0.6923 |
|  | Multi | 1.0402 | 0.6894 |
| TTM | Uni | 1.0400 | 0.6948 |
|  | Multi | 1.0238 | 0.6893 |
| CRU | Uni | 1.0282 | 0.6968 |
|  | Multi | 1.0267 | 0.6912 |
| LatentODE | Uni | 1.0271 | 0.6886 |
|  | Multi | 1.0268 | 0.6856 |
| NeuralFlow | Uni | 1.0265 | 0.6942 |
|  | Multi | 1.0260 | 0.6958 |
| tPatchGNN | Uni | 1.0267 | 0.6893 |
|  | Multi | 1.0265 | 0.6958 |

Table 4: Unimodal vs. multimodal forecasting results on the RepoHealth dataset.

| Model | Modal | MSE | MAE |
|---|---|---|---|
| Informer | Uni | 0.6391 | 0.4143 |
| | Multi | 0.6296 | 0.4037 |
| DLinear | Uni | 0.5376 | 0.4081 |
| | Multi | 0.5291 | 0.4021 |
| PatchTST | Uni | 0.5727 | 0.4185 |
| | Multi | 0.5520 | 0.4180 |
| TimesNet | Uni | 0.5415 | 0.4158 |
| | Multi | 0.5599 | 0.4275 |
| TimeMixer | Uni | 0.7267 | 0.4321 |
| | Multi | 0.6890 | 0.4230 |
| TimeLLM | Uni | 0.5655 | 0.3994 |
| | Multi | 0.5601 | 0.3919 |
| TTM | Uni | 0.5442 | 0.4088 |
| | Multi | 0.5386 | 0.4079 |
| CRU | Uni | 1.0362 | 0.8030 |
| | Multi | 0.8538 | 0.6596 |
| LatentODE | Uni | 0.7555 | 0.6104 |
| | Multi | 0.7341 | 0.6285 |
| NeuralFlow | Uni | 0.8231 | 0.6736 |
| | Multi | 0.7720 | 0.6409 |
| tPatchGNN | Uni | 0.6196 | 0.5000 |
| | Multi | 0.6090 | 0.4718 |

Table 5: Unimodal vs. multimodal forecasting results on the MIMIC dataset.

| Model | Modal | MSE | MAE |
|---|---|---|---|
| Informer | Uni | 0.8519 | 0.5588 |
| | Multi | 0.7988 | 0.6083 |
| DLinear | Uni | 0.8770 | 0.6152 |
| | Multi | 0.8515 | 0.6456 |
| PatchTST | Uni | 0.9394 | 0.6794 |
| | Multi | 0.9281 | 0.6799 |
| TimesNet | Uni | 1.0374 | 0.7348 |
| | Multi | 0.9335 | 0.6821 |
| TimeMixer | Uni | 0.8856 | 0.5686 |
| | Multi | 0.8179 | 0.5821 |
| TimeLLM | Uni | 0.7901 | 0.5457 |
| | Multi | 0.7535 | 0.5620 |
| TTM | Uni | 0.8620 | 0.6283 |
| | Multi | 0.9105 | 0.6707 |
| CRU | Uni | 1.0122 | 0.7235 |
| | Multi | 0.9847 | 0.7087 |
| LatentODE | Uni | 0.9325 | 0.6868 |
| | Multi | 0.9423 | 0.6785 |
| NeuralFlow | Uni | 1.0086 | 0.7219 |
| | Multi | 0.9884 | 0.7079 |
| tPatchGNN | Uni | 0.7250 | 0.5479 |
| | Multi | 0.7656 | 0.5794 |

Table 6: Unimodal vs. multimodal forecasting results on the FNSPID dataset.

| Model | Modal | MSE | MAE |
| --- | --- | --- | --- |
| Informer | Uni | 0.1277 | 0.2153 |
| | Multi | 0.1282 | 0.2149 |
| DLinear | Uni | 0.1239 | 0.2121 |
| | Multi | 0.1248 | 0.2145 |
| PatchTST | Uni | 0.1301 | 0.2240 |
| | Multi | 0.1277 | 0.2212 |
| TimesNet | Uni | 0.1253 | 0.2144 |
| | Multi | 0.1239 | 0.2128 |
| TimeMixer | Uni | 0.1161 | 0.1965 |
| | Multi | 0.1157 | 0.1964 |
| TimeLLM | Uni | 0.1261 | 0.2138 |
| | Multi | 0.1240 | 0.2124 |
| TTM | Uni | 0.1355 | 0.2373 |
| | Multi | 0.1335 | 0.2312 |
| CRU | Uni | 0.1484 | 0.2545 |
| | Multi | 0.1476 | 0.2471 |
| LatentODE | Uni | 0.1203 | 0.2134 |
| | Multi | 0.1157 | 0.2051 |
| NeuralFlow | Uni | 0.1510 | 0.2572 |
| | Multi | 0.1420 | 0.2470 |
| tPatchGNN | Uni | 0.1247 | 0.2195 |
| | Multi | 0.1212 | 0.2132 |

Table 7: Unimodal vs. multimodal forecasting results on the ClusterTrace dataset.

| Model | Modal | MSE | MAE |
|---|---|---|---|
| Informer | Uni | 1.0793 | 0.8039 |
| | Multi | 0.7473 | 0.6527 |
| DLinear | Uni | 0.8418 | 0.6765 |
| | Multi | 0.6440 | 0.6504 |
| PatchTST | Uni | 2.0366 | 0.9896 |
| | Multi | 1.0370 | 0.8389 |
| TimesNet | Uni | 0.9739 | 0.7187 |
| | Multi | 0.6706 | 0.6579 |
| TimeMixer | Uni | 1.1262 | 0.7861 |
| | Multi | 0.7377 | 0.6946 |
| TimeLLM | Uni | 0.9596 | 0.7325 |
| | Multi | 0.7250 | 0.6973 |
| TTM | Uni | 1.0030 | 0.7409 |
| | Multi | 0.7166 | 0.6813 |
| CRU | Uni | 1.2111 | 0.8287 |
| | Multi | 0.5753 | 0.5988 |
| LatentODE | Uni | 1.1613 | 0.7818 |
| | Multi | 0.5336 | 0.5780 |
| NeuralFlow | Uni | 1.4458 | 0.8910 |
| | Multi | 0.6248 | 0.6384 |
| tPatchGNN | Uni | 0.9671 | 0.7119 |
| | Multi | 0.6105 | 0.6104 |

Table 8: Unimodal vs. multimodal forecasting results on the StudentLife dataset.

| Model | Modal | MSE | MAE |
|---|---|---|---|
| Informer | Uni | 0.9432 | 0.6674 |
| | Multi | 0.8910 | 0.6790 |
| DLinear | Uni | 0.8850 | 0.6760 |
| | Multi | 0.8733 | 0.6629 |
| PatchTST | Uni | 0.8964 | 0.6793 |
| | Multi | 0.8764 | 0.6814 |
| TimesNet | Uni | 0.9069 | 0.6827 |
| | Multi | 0.8892 | 0.6889 |
| TimeMixer | Uni | 0.9437 | 0.6720 |
| | Multi | 0.8867 | 0.6768 |
| TimeLLM | Uni | 0.9355 | 0.6797 |
| | Multi | 0.8870 | 0.6781 |
| TTM | Uni | 0.9024 | 0.6830 |
| | Multi | 0.8888 | 0.6838 |
| CRU | Uni | 0.9101 | 0.6876 |
| | Multi | 0.9101 | 0.6890 |
| LatentODE | Uni | 0.9028 | 0.6965 |
| | Multi | 0.9040 | 0.6907 |
| NeuralFlow | Uni | 0.9237 | 0.6995 |
| | Multi | 0.9150 | 0.7024 |
| tPatchGNN | Uni | 0.8661 | 0.6672 |
| | Multi | 0.8633 | 0.6705 |

Table 9: Unimodal vs. multimodal forecasting results on the ILINet dataset.

| Model | Modal | MSE | MAE |
|---|---|---|---|
| Informer | Uni | 1.2714 | 0.8118 |
|  | Multi | 1.2338 | 0.7832 |
| DLinear | Uni | 1.0402 | 0.6811 |
|  | Multi | 0.9795 | 0.6374 |
| PatchTST | Uni | 1.1606 | 0.7532 |
|  | Multi | 1.1727 | 0.7373 |
| TimesNet | Uni | 1.0518 | 0.6670 |
|  | Multi | 1.0510 | 0.6436 |
| TimeMixer | Uni | 0.9825 | 0.6456 |
|  | Multi | 1.0530 | 0.6845 |
| TimeLLM | Uni | 1.1243 | 0.7255 |
|  | Multi | 1.0333 | 0.6901 |
| TTM | Uni | 1.1010 | 0.7003 |
|  | Multi | 1.0383 | 0.6898 |
| CRU | Uni | 0.9718 | 0.6652 |
|  | Multi | 1.0423 | 0.6900 |
| LatentODE | Uni | 1.1512 | 0.7531 |
|  | Multi | 1.1098 | 0.7276 |
| NeuralFlow | Uni | 1.1946 | 0.7688 |
|  | Multi | 1.1501 | 0.7460 |
| tPatchGNN | Uni | 1.6163 | 0.9255 |
|  | Multi | 1.4877 | 0.8689 |

Table 10: Unimodal vs. multimodal forecasting results on the CESNET dataset.

| Model | Modal | MSE | MAE |
| --- | --- | --- | --- |
| Informer | Uni | 0.9591 | 0.7559 |
| | Multi | 0.8754 | 0.7233 |
| DLinear | Uni | 0.9695 | 0.7517 |
| | Multi | 0.8763 | 0.7221 |
| PatchTST | Uni | 1.3177 | 0.8797 |
| | Multi | 1.1242 | 0.8192 |
| TimesNet | Uni | 0.9581 | 0.7499 |
| | Multi | 0.8686 | 0.7201 |
| TimeMixer | Uni | 0.9500 | 0.7481 |
| | Multi | 0.8588 | 0.7160 |
| TimeLLM | Uni | 0.9891 | 0.7676 |
| | Multi | 0.9125 | 0.7364 |
| TTM | Uni | 0.9869 | 0.7653 |
| | Multi | 0.9205 | 0.7422 |
| CRU | Uni | 0.9462 | 0.7559 |
| | Multi | 0.8726 | 0.7257 |
| LatentODE | Uni | 0.9655 | 0.7642 |
| | Multi | 0.8977 | 0.7400 |
| NeuralFlow | Uni | 1.0154 | 0.7915 |
| | Multi | 0.9495 | 0.7685 |
| tPatchGNN | Uni | 0.9796 | 0.7713 |
| | Multi | 0.9122 | 0.7483 |

Table 11: Unimodal vs. multimodal forecasting results on the EPA-Air dataset.

| Model | Modal | MSE | MAE |
| --- | --- | --- | --- |
| Informer | Uni | 0.6301 | 0.5983 |
| | Multi | 0.5812 | 0.5728 |
| DLinear | Uni | 0.5361 | 0.5279 |
| | Multi | 0.5223 | 0.5223 |
| PatchTST | Uni | 0.6196 | 0.5947 |
| | Multi | 0.6204 | 0.5992 |
| TimesNet | Uni | 0.5599 | 0.5524 |
| | Multi | 0.5892 | 0.5650 |
| TimeMixer | Uni | 0.6086 | 0.5770 |
| | Multi | 0.5641 | 0.5663 |
| TimeLLM | Uni | 0.5835 | 0.5643 |
| | Multi | 0.5334 | 0.5303 |
| TTM | Uni | 0.6002 | 0.5653 |
| | Multi | 0.6218 | 0.5761 |
| CRU | Uni | 0.7026 | 0.6306 |
| | Multi | 0.7982 | 0.6739 |
| LatentODE | Uni | 0.8025 | 0.6665 |
| | Multi | 0.7556 | 0.6523 |
| NeuralFlow | Uni | 0.7821 | 0.6488 |
| | Multi | 0.8202 | 0.6790 |
| tPatchGNN | Uni | 0.6258 | 0.6022 |
| | Multi | 0.5840 | 0.5793 |

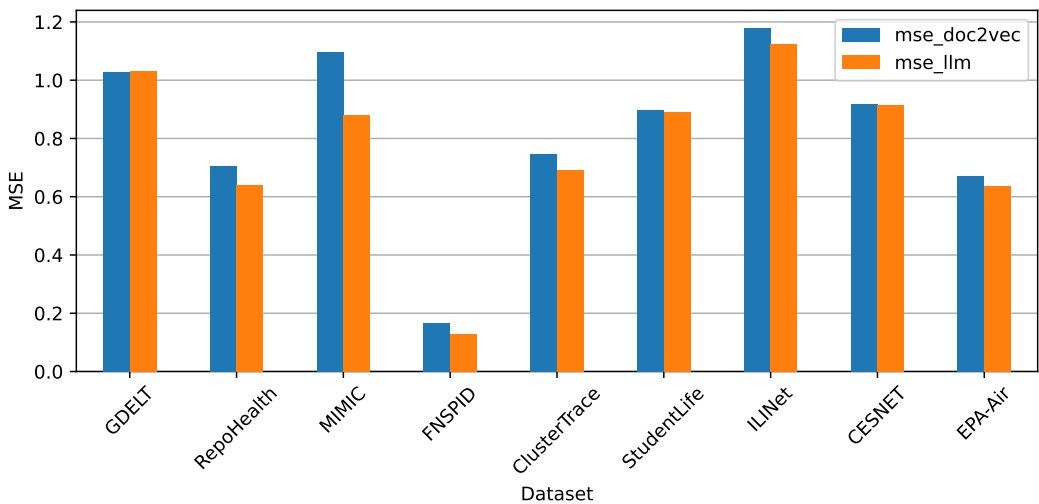

Figure 10: Performance comparison between frozen LLM-based text encoders and Doc2Vec, averaged across all 11 baselines for each dataset.

# M   Text Encoder Variants

This section investigates how different text encoders affect multimodal forecasting performance. We consider two contrasting setups:

- **Lightweight Encoder: Doc2Vec.** A classical shallow embedding model trained from scratch on the dataset corpus.

- **Large-Scale LLM Encoder: Llama-70B.** A state-of-the-art large language model used in a frozen setting to generate contextual embeddings.

## M.1   Doc2Vec vs. Frozen LLMs

To evaluate the importance of pretrained semantics, we replace frozen LLM-based text encoders with Doc2Vec, which learns fixed-length embeddings directly on each dataset's text corpus. Figure 10 summarizes results averaged across all datasets and baselines.

Models using pretrained LLM embeddings consistently outperform those using Doc2Vec, confirming the importance of semantic knowledge from large-scale pretraining—even under irregular and domain-specific text distributions.

## M.2   Scaling to Very Large LLMs (Llama-70B)

We also investigate whether scaling up the text encoder leads to further improvements by evaluating a frozen **Llama-70B** variant. However, larger models were excluded from full-scale experiments due to their extremely high memory demands ($\geq$48 GB) and limited added value. To validate this decision, we compared several representative text encoders—including GPT-2, BERT, Llama-8B, Llama-70B, and DeepSeek-7B—on three datasets (GDELT, RepoHealth, and MIMIC).

As shown in Table 12, Llama-70B consistently underperforms its smaller 8B counterpart across all tested datasets, offering no clear advantage despite significantly higher computational cost. These results support our decision to focus on smaller-scale encoders that offer a better trade-off between efficiency and accuracy. Nonetheless, IMM-TSF remains fully modular, allowing users to substitute larger encoders through a simple configuration change if desired.

Table 12: Performance comparison of different text encoders on selected datasets. Llama-70B provides no clear benefit over smaller models despite substantially higher computational cost.

| Dataset | GPT-2 | BERT | Llama-8B | Llama-70B | DeepSeek-7B |
|---------|-------|------|----------|-----------|-------------|
| GDELT | **1.066** | **1.066** | **1.066** | 1.069 | 1.067 |
| RepoHealth | **0.540** | 0.541 | 0.545 | 0.547 | 0.545 |
| MIMIC | 0.514 | 0.571 | **0.491** | 0.596 | 0.524 |

Table 13: Efficiency comparison of different TTF and MMF module combinations on the RepoHealth dataset. Training time is measured per epoch, and inference time per forward pass (averaged over 1,000 samples).

| TTF + MMF Module | # Params | Training Time (s) | Inference Time (ms) | MSE |
|------------------|----------|-------------------|---------------------|-----|
| $\text{TTF}_{\text{RecAvg}}$ + $\text{MMF}_{\text{GR-Add}}$ | 1.20M | 2.91 | 80.05 | **0.540** |
| $\text{TTF}_{\text{RecAvg}}$ + $\text{MMF}_{\text{XAttn-Add}}$ | 4.73M | 3.94 | 99.20 | 0.687 |
| $\text{TTF}_{\text{T2V-XAttn}}$ + $\text{MMF}_{\text{GR-Add}}$ | 4.45M | 4.97 | 131.26 | 0.546 |
| $\text{TTF}_{\text{T2V-XAttn}}$ + $\text{MMF}_{\text{XAttn-Add}}$ | 7.98M | 5.03 | 154.34 | 0.683 |

# N    Computational Cost and Efficiency Analysis

This appendix provides a detailed analysis of the computational efficiency of IMM-TSF, focusing on both fusion module design and text encoder selection. We report parameter counts, training and inference time, and forecasting performance (MSE) to highlight the trade-offs between model complexity and accuracy.

## N.1    Fusion Module Efficiency

Both Temporal-Text Fusion (TTF) modules have a fixed number of parameters, while Multimodal Fusion (MMF) modules scale with the number of time-series features. To illustrate these trade-offs, Table 13 reports the number of parameters, training time, inference time, and forecasting performance on the RepoHealth dataset, which contains 10 time-series features.

For the TTF component, both *RecAvg* and *T2V-XAttn* achieve comparable accuracy, suggesting that simple recency-based averaging is sufficient for temporal-text alignment. For the MMF component, *GR-Add* outperforms *XAttn-Add*, likely due to its gating mechanism that effectively filters out noisy or redundant text features. Overall, the combination $\text{TTF}_{\text{RecAvg}}$ + $\text{MMF}_{\text{GR-Add}}$ offers the best balance of speed and accuracy, making it a strong default configuration for most applications.

## N.2    Text Encoder Efficiency

All textual embeddings are projected into a shared 768-dimensional latent space using a learned linear transformation before multimodal fusion. Since the text encoders are frozen, we precompute document embeddings once and reuse them throughout training, which substantially reduces runtime overhead.

Table 14 compares the computational cost of generating embeddings across various text encoders. The embedding time represents the average processing time per document on a single GPU.

While larger encoders incur higher one-time embedding costs, all models remain practical in our setup due to precomputation. For real-time or resource-constrained deployment, GPT-2 and BERT

Table 14: Comparison of text encoder efficiency. All embeddings are precomputed before training; MSE is measured on the RepoHealth dataset.

| Text Encoder | # Params | Embedding Time (s) | MSE |
|---|---|---|---|
| GPT-2 | 124.44M | 0.04 | **0.540** |
| BERT | 109.48M | **0.02** | 0.541 |
| DeepSeek-7B | 6,481.17M | 1.02 | 0.545 |
| Llama-8B | 7,504.92M | 1.06 | 0.547 |
| Llama-70B | 69,503.03M | 17.16 | 0.545 |

Table 15: Forecasting performance on the GDELT dataset using different auxiliary modalities. All configurations use the same fusion and forecasting modules, with only the encoder swapped.

| Auxiliary Modality | MSE |
|---|---|
| None | 1.067 |
| Text | **1.066** |
| Image | 1.675 |

provide the best efficiency–accuracy trade-off. In contrast, large-scale encoders such as Llama-70B and DeepSeek-7B offer minimal accuracy gains relative to their significant computational overhead.

# O   Extending IMM-TSF Beyond Textual Modalities

While our initial release focuses on text as the auxiliary modality, IMM-TSF is designed to be modality-agnostic and easily extensible to other asynchronous data types such as images or audio. This design choice aligns with our discussion in the *"Beyond Textual Modalities"* paragraph of the Future Work section, where we emphasize the framework's goal as a flexible foundation for multimodal irregular time series forecasting.

## O.1   Adding an Image Modality to GDELT

To empirically validate this extensibility, we augment the GDELT dataset with an additional image modality. We collect the images embedded within each GDELT news article and encode them using a pretrained Vision Transformer (ViT). Importantly, this integration required only replacing the LLM-based text encoder with an image encoder, without modifying the remaining components of the IMM-TSF pipeline. This minimal change demonstrates the plug-and-play nature of our architecture.

As shown in Table 15, the image-only variant underperforms text in this setting, likely due to weaker semantic alignment between visual content and the target financial time series. Nevertheless, this experiment confirms that IMM-TSF seamlessly supports alternative modalities with minimal implementation effort, enabling future exploration of cross-modal irregularity at scale.

## O.2   Implementation Example

The following code snippet illustrates how the image modality was integrated into the existing IMM-TSF pipeline. The only required change is swapping the encoder used for the auxiliary input:

```
# Original text-based encoder (frozen LLM)
notes_input = encode_text(raw_input, model=text_encoder)

# Replaced with image encoder (no other changes to fusion pipeline)
notes_input = encode_image(raw_input, model=image_encoder)
```

This simple substitution highlights one of IMM-TSF's core strengths: its modality-agnostic archi-tecture, which decouples the encoder from the fusion and forecasting components. By abstracting modality-specific processing within a unified interface, IMM-TSF enables benchmarking, fusion, and forecasting across diverse asynchronous input types with minimal code changes.

We will release this image-augmented GDELT variant as part of a follow-up to the TIME-IMM benchmark suite, providing a concrete example of how researchers can extend IMM-TSF to additional modalities such as audio, images, or structured event streams.

## P    Temporal Generalization Analysis

To evaluate whether models trained on TIME-IMM generalize across time rather than merely across samples, we conducted a rolling-origin cross-validation experiment. While Section 4.1 already describes a strict chronological split (train < validation < test), this analysis further tests stability under temporal distribution shifts.

We performed 5-fold rolling-origin cross-validation, where the forecasting origin advances forward in time with each fold. In each fold, the model is trained on an expanding historical window and evaluated on the immediately subsequent period, mimicking deployment in evolving real-world environments. This setup measures robustness to time shifts in the raw input distribution.

Table 16: Rolling-origin cross-validation (5 folds) evaluating temporal generalization on TIME-IMM. MSE and coefficient of variation (CV) are reported for each dataset.

| Dataset | Fold 1 | Fold 2 | Fold 3 | Fold 4 | Fold 5 | Mean MSE | MSE CV (%) |
|---|---|---|---|---|---|---|---|
| GDELT | 0.9183 | 0.9512 | 0.9654 | 1.1515 | 0.9768 | 0.9926 | 9.22 |
| RepoHealth | 0.4136 | 0.5210 | 0.5001 | 0.5086 | 0.5951 | 0.5077 | 12.74 |
| MIMIC | 1.2220 | 1.1000 | 0.9131 | 1.0180 | 0.9240 | 1.0354 | 12.47 |
| FNSPID | 0.1584 | 0.1327 | 0.1219 | 0.1766 | 0.1456 | 0.1470 | 14.60 |
| ClusterTrace | 0.9815 | 0.9626 | 0.6759 | 1.0628 | 0.7351 | 0.8836 | 19.03 |
| StudentLife | 0.9284 | 0.8954 | 0.8989 | 0.8397 | 0.8803 | 0.8885 | 3.65 |
| ILINet | 0.8491 | 0.7179 | 0.6453 | 0.5740 | 0.7492 | 0.7071 | 14.76 |
| CESNET | 1.1981 | 1.0456 | 1.3465 | 0.8689 | 0.9626 | 1.0843 | 17.52 |
| EPA-Air | 0.6492 | 0.6340 | 0.5664 | 0.7883 | 0.5186 | 0.6313 | 16.21 |

Across all datasets, the average coefficient of variation (CV) for MSE is 13.35%, with every dataset remaining below 20%. These results indicate that models trained on TIME-IMM maintain stable forecasting accuracy under temporal drift, demonstrating strong generalization across different time windows.

