# OpenReview forum: "Time-IMM: A Dataset and Benchmark for Irregular Multimodal Multivariate Time Series"
_NeurIPS.cc/2025/Datasets_and_Benchmarks_Track — NeurIPS 2025 Datasets and Benchmarks Track poster_

### Official Review · Reviewer_6rkv · 2025-06-22

**Rating:** 4
**Confidence:** 4

**Summary:**

The paper introduces a Time-IMM, a datasets with irregularly sampled patterns for multi-modality time series with text information. A plug-and-play library named IMM-TSF is also provided for forecasting tasks on irregular multi-modal time series data. The experiments demonstrate that model with textual data using multiple fusion modules can perform better in the time series forecasting tasks.

**Additional Feedback:**

**Questions**

1. Have the authors tried smaller text encoders such as DistillBert during experiments? Did that influence the performance a lot?
2. Have the authors tested if the GPT summarize actually avoid hallucination and extract the important facts? How is the dataset constructed? What evaluation methods are used?
3. Have the authors tested if the model trained on Time-IMM performs good enough on other raw data of different time frame from original source? This might be helpful to ensure the quality of the dataset.

**Dataset Code Accessibility:**

Yes

**Dataset Code Comments:**

The authors provide easy access to the datasets using Kaggle platform, and the source codes for the plug-in-play library mentioned in the paper is given as well. It is good to see that there are a lot of different layers / baseline models are provided as well, which makes it easy for following researches to reproduce baseline results based on their own settings. Besides, the authors provide good base class to extend different fusion models which also is accessible for researchers to develop new fusion algorithms for the multi-modal time series modeling.

**Ethical Considerations:**

No, there are no or only very minor ethics concerns

**Final Justification:**

The authors provide good response to cover my concerns on some of the experiment results. I recommend accept of the paper.

**Limitations Weaknesses:**

1. Use LLM to summarize text might bring hallucination and result in text data following implicit patterns. Not clear why need to summarize instead of using original texts. Better to have small ablation experiments about this aspect.
2. It appears in the experiments (e.g., figure 6 (a) and figure 6 (c)) that texts are not super helpful for forecasting for certain datasets. Besides, different choices of pretrained text encoders does not demonstrate huge difference of performances. Wonder whether texts are actually fully utilized since BERT is significantly smaller than LLaMA 3.1 and DeepSeek.
3. Each dataset only represents one patten of irregularity. It would be better if there are datasets that have multiple different patterns which pose a more challenging task and mimic other real-world scenarios. It is recommended the authors can provide some insights on these and propose methods / APIs to obtain such datasets as well.

**Strengths Contributions:**

1. The paper is well-written with clear structures, images, tables, etc. It is easy to follow.
2. The paper has good motivation compared with existing datasets. It introduces time series datasets with multiple causes for irregular patterns, and also incorporate multi-modality data for forecasting at the same time.
3. A plug-in-play python library is provided for future research, which contains unimodal time series forecasting model, text encoders, timestamp-to-text fusion module, multi-modal fusion module as well as training / data loading / evaluation modules.

---

> ### Author Rebuttal · Authors · 2025-07-30
>
> > ### W1: Justification and validation of LLM-based text summarization (also Q2)
>
> We use summarized textual data instead of the original text because most of the included datasets are proprietary and cannot be released under NeurIPS’s CC BY 4.0 requirement. This approach follows common practice for releasing real-world datasets, such as Time-MMD. To ensure the quality and topic accuracy of our dataset, we performed a rigorous validation of our GPT-4.1 Nano-based pipeline by manually verifying 250 summaries. Our evaluation shows that the model identifies relevant documents with 100% recall (ensuring no relevant data is missed) and 88.4% precision.
>
> ---
>
> > ### W2: Effectiveness and utilization of textual modality in forecasting
>
> Textual impact varies by dataset due to differences in semantic informativeness. For example, FNSPID (financial news + stock prices) shows smaller gains because stock trends are already strongly expressed in the numerical series, and associated text often provides indirect or lagging context. In contrast, ClusterTrace benefits more significantly from textual data, as its workload annotations often describe operational changes that directly affect future values. These observations indicate that performance gains are driven more by the semantic relevance and predictive strength of the text, rather than the irregularity type alone.
>
> Regarding the insensitivity to language model size, we believe this stems from two factors. First, irregular TSF hinges more on temporal alignment and contextual anchoring than on deep semantic reasoning, making larger LLMs less essential. Second, although our TTF and MMF modules enable effective integration of textual signals, the scale of textual input remains relatively small—most summaries are distilled into fewer than five sentences per time step. This is orders of magnitude smaller than the training corpora for LLaMA3.1 or DeepSeek, and thus may not fully leverage the representational capacity of such large models.
>
> Empirically, all datasets consistently benefit from the addition of text. This validates that the textual modality is being effectively utilized, even if the marginal gain varies depending on task semantics and dataset characteristics.
>
> ---
>
> > ### W3: Toward multi-irregularity datasets and realistic benchmarking
>
> In this work, we design our benchmark around atomic irregularity types, with each dataset representing a clearly defined and interpretable pattern. This deliberate choice enables precise analysis of how different irregularity types affect forecasting and allows us to build a taxonomy-grounded foundation for future modeling and evaluation efforts.
>
> We agree that datasets combining multiple irregularity sources pose a more realistic and challenging benchmark scenario. As a next step, we plan to support compositional irregularity generation, enabling the construction of hybrid datasets by combining different atomic patterns. We are developing APIs that allow users to specify irregularity combinations (e.g., human scheduling + missing data) and generate synthetic or semi-synthetic datasets that reflect more complex real-world dynamics. We will expand the discussion in the paper to include this future direction and thank the reviewer for highlighting this important aspect.
>
> ---
>
> > ### Q1: Effect of DistilBERT or other lightweight encoders?
>
> We have already included results for a lightweight, train-from-scratch encoder (Doc2Vec) in Appendix L.1. Across the board, models using frozen pretrained LLMs consistently outperform Doc2Vec, underscoring the value of semantic priors even in asynchronous, domain-specific settings.
>
> To broaden this comparison, we also evaluate DistilBERT, a smaller pretrained model that offers better efficiency. The table below compares forecasting performance (MSE) across Doc2Vec, BERT, and DistilBERT on all nine datasets:
>
> | Dataset          | mse_doc2vec | mse_bert | mse_distilbert |
> | ---------------- | ------------- | ---------- | ---------------- |
> | GDELT            | **1.029**     | 1.047    | *1.034*            |
> | RepoHealth       | 0.625         | **0.547**  | *0.614*          |
> | MIMIC            | 1.317         | *0.991*    | **0.963**        |
> | FNSPID           | 0.134         | **0.129**  | *0.133*          |
> | ClusterTrace     | *0.949*         | **0.844**  | 1.120            |
> | StudentLife      | 0.885         | *0.885*    | **0.879**        |
> | ILINet           | **1.064**     | *1.131*    | 1.173            |
> | CESNET           | *0.949*     | 0.970      | **0.944**          |
> | EPA-Air          | **0.610**     | *0.617*    | 0.624            |
> | **Average Rank** | 2.11      | **1.89**     | *2.00*             |
>
> In our comparison across **Doc2Vec**, **BERT**, and **DistilBERT**, we observe the following trends:
>
> * **BERT** is the most stable performer, never ranking last and maintaining strong generalization across domains.
> * **DistilBERT** achieves comparable or better accuracy in many settings while reducing computational cost, making it ideal for efficiency-constrained deployments.
> * **Doc2Vec**, despite its simplicity, can outperform both in repetitive, large-scale reporting scenarios where pretrained contextual understanding is less critical.
>
> These results further highlight that model choice should consider the nature of the text and task, not just model capacity.
>
> ---
>
> > ### Q3: Does a model trained on Time-IMM generalise to raw data from different time windows?
>
> Time-IMM already employs a strict chronological data split (*train < val < test*) to reflect realistic forecasting deployments (Section 4.1). To further assess temporal generalization, we conducted a **5-fold rolling-origin cross-validation**, where the forecasting “origin” (i.e., the prediction start point) advances forward in time with each fold. In each fold, the model is trained on an expanding historical window and tested on the immediately subsequent block, mimicking how models are applied to unseen future periods in deployment settings. This protocol tests generalization not just across samples, but across **time shifts** in the raw input distribution.
>
> The results of this analysis are presented in the table below:
>
> | Dataset      | Fold 1 | Fold 2 | Fold 3 | Fold 4 | Fold 5 | Mean MSE | MSE CV (%) |
> | ------------ | ------ | ------ | ------ | ------ | ------ | -------- | ---------- |
> | GDELT        | 0.9183 | 0.9512 | 0.9654 | 1.1515 | 0.9768 | 0.9926   | 9.22%      |
> | RepoHealth   | 0.4136 | 0.521  | 0.5001 | 0.5086 | 0.5951 | 0.5077   | 12.74%     |
> | MIMIC        | 1.222  | 1.1    | 0.9131 | 1.018  | 0.924  | 1.0354   | 12.47%     |
> | FNSPID       | 0.1584 | 0.1327 | 0.1219 | 0.1766 | 0.1456 | 0.1470   | 14.60%     |
> | ClusterTrace | 0.9815 | 0.9626 | 0.6759 | 1.0628 | 0.7351 | 0.8836   | 19.03%     |
> | StudentLife  | 0.9284 | 0.8954 | 0.8989 | 0.8397 | 0.8803 | 0.8885   | 3.65%      |
> | ILINet       | 0.8491 | 0.7179 | 0.6453 | 0.574  | 0.7492 | 0.7071   | 14.76%     |
> | CESNET       | 1.1981 | 1.0456 | 1.3465 | 0.8689 | 0.9626 | 1.0843   | 17.52%     |
> | EPA-Air      | 0.6492 | 0.634  | 0.5664 | 0.7883 | 0.5186 | 0.6313   | 16.21%     |
>
> Across all datasets, the average coefficient of variation (CV) for MSE is 13.35%, with all values remaining below 20%. This indicates **robust generalization over time** with minimal performance degradation, even as temporal drift accumulates.

---

> > ### Comment · Reviewer_6rkv · 2025-08-06
> > **Response**
> >
> > Thank you for the detailed response. I will maintain the scores. Please also make sure to add these additional results in the final version.

---

### Official Review · Reviewer_6XNE · 2025-06-23

**Rating:** 5
**Confidence:** 4

**Summary:**

This paper proposes Time-IMM, a cause-driven irregular multimodal dataset, and IMM-TSF, a benchmark library for irregular multimodal time series forecasting. Specifically, this paper classifies the irregularities into 3 categories and 9 types, and constructed one dataset for each type. This paper also shows in the experiments that incorporating multimodality could improve the performance for irregular time series forecasting.

**Dataset Code Accessibility:**

Yes

**Dataset Code Comments:**

Annoymized GitHub repo link provided in the paper, dataset also available in Kaggle.

**Ethical Considerations:**

No, there are no or only very minor ethics concerns

**Final Justification:**

Thanks for the response from the author. The reason that "there's no principled way to define a “match” between a given text and specific time steps." makes sense to me, and I will also update the score.

**Limitations Weaknesses:**

My main concern for the paper is that the texts are assigned with independent timestamps, and no alignment is assumed between the text and the time series. Although I understand the authors' concern that texts may arrive asynchronously and may not be temporally aligned with the timestamps of numerical time series, at least high-level matching during fusion could still be helpful. This could also leave the decision on whether to use the text-to-time series matching as an option to users.

**Strengths Contributions:**

1. Good motivation: Inregualrity in time series is one of the challenging problem in time series forecasting, and few works focus on using multi-modal frameworks to tackle this problem.
2. Good experiment results: Experiment results and ablation studies showcase that incorporating multimodality do improve the performance for irregular time series forecasting.
3. The dataset and code repo are well-structured, clearly wrttien with easy-to-understand readme files.

---

> ### Author Rebuttal · Authors · 2025-07-30
>
> > ### W1: Lack of explicit alignment between text and time series during fusion
>
> We appreciate the reviewer’s thoughtful concern. While the textual and numerical data in Time-IMM are asynchronously timestamped, we **intentionally avoid hard alignment during fusion** for the following reasons:
>
> 1. **Alignment is ambiguous in irregular settings.**
>    Time-IMM datasets involve unevenly sampled time series with missing values and asynchronous textual updates. In many cases, there's no principled way to define a “match” between a given text and specific time steps. Forcing an alignment could result in dropped signals or introduce incorrect associations.
>
> 2. **Our fusion module handles temporal context implicitly.**
>    Instead of aligning text and time series explicitly, our **timestamp-to-text fusion (TTF) module** (Section 3.2) constructs temporally-aware representations by incorporating the relative time distance between text and forecast queries. This allows the model to **learn soft associations** between modalities based on timestamp offsets, without requiring exact alignment.
>
> 3. **We preserve user flexibility for downstream adaptation.**
>    By modeling the data in its naturally asynchronous form, we enable future work to explore alternate alignment strategies (e.g., window-based matching, attention gating, or delay modeling). Our framework does not preclude these options—it simply does not impose them upfront.
>
> We will clarify this rationale in the revised manuscript and thank the reviewer for raising an important modeling design consideration.

---

> ### Comment · Area_Chair_1ntq · 2025-08-06
>
> Dear Reviewer,
>
> Thank you for your valuable feedback. The authors have addressed your comments in their rebuttal. We kindly ask that you engage in discussion with the authors before submitting your Mandatory Acknowledgement.
>
> If your concerns have been adequately addressed in the rebuttal, please let the authors know. If your concerns remain unresolved, please communicate that clearly as well.
>
> Thank you for contributing to a fair and constructive review process at NeurIPS.

---

### Official Review · Reviewer_4Has · 2025-07-02

**Rating:** 4
**Confidence:** 3

**Summary:**

The paper introduces Time-IMM, a benchmark dataset designed to address the challenges of irregular, multimodal, multivariate time series data. The dataset categorizes irregularities into nine types, including trigger-based, constraint-based, and artifact-based mechanisms. Accompanying the dataset is the IMM-TSF benchmark library, which supports forecasting on irregular multimodal time series through specialized fusion modules. The paper demonstrates that modeling multimodality in irregular time series data significantly improves forecasting performance. The findings underscore the importance of multimodal modeling in improving forecasting accuracy under real-world conditions.

**Dataset Code Accessibility:**

Yes

**Ethical Considerations:**

No, there are no or only very minor ethics concerns

**Final Justification:**

As authors have included additional modality and described the key differences from existing datasets, I will this work is worth publishing.

**Limitations Weaknesses:**

I think the paper provides limited value to the community, as these datasets are existing datasets and it simply processes and aggregates them.
The focus on textual data as the secondary modality may limit the applicability of the findings to scenarios involving other types of asynchronous inputs, such as images or audio.

**Strengths Contributions:**

1: It categorizes real-world irregularities into nine cause-driven types, providing a comprehensive resource for studying irregular time series. Such categorization of irregularities provides a valuable framework for understanding and modeling real-world time series data.
2: IMM-TSF is introduced as a modular library for forecasting, supporting asynchronous integration of numerical and textual data. This offers a flexible platform for experimenting with different forecasting strategies, promoting robust research in the field.
3: The paper shows that incorporating multimodality in irregular time series modeling leads to substantial improvements in forecasting accuracy.

---

> ### Author Rebuttal · Authors · 2025-07-30
>
> > ### W1: The benchmark "simply processes and aggregates existing datasets," offering little new data to the community.
>
> We understand the concern and would like to clarify that Time-IMM involves **significant data construction, multimodal integration, and targeted curation**, far beyond simple aggregation.
>
> Three of the nine datasets—**RepoHealth, CESNET, and EPA-Air**—were **collected entirely from scratch**, involving original data scraping, preprocessing, and fusion of numerical and textual streams. For the remaining datasets, we undertook substantial work to convert raw or loosely structured resources into coherent, temporally aligned, multimodal forecasting tasks designed around specific irregularity types:
>
> * **MIMIC**: While widely used, MIMIC lacks an established benchmark that pairs clinical notes and vitals for long-range multimodal forecasting. We built this variant from raw EHR data, ensuring sufficient temporal coverage and alignment for forecasting use.
>
> * **ClusterTrace**: Existing work analyzes resource usage without job context and flattens its temporal complexity. We transformed ClusterTrace by preserving its complex temporal dynamics (concurrent jobs, DAG dependencies) and adding contextual job descriptions, enabling context-aware resource forecasting.
>
> * **StudentLife**: Previous studies analyze sensor data apart from psychological assessments. Our work enables linking emotional states to future behaviors by converting standardized scales into behavioral language while harmonizing cross-platform sensors.
>
> * **GDELT**: The original data provides only URLs. We scraped the full text of articles, then applied a domain-specific GPT-4.1 Nano summarization pipeline to filter and compress content with minimal hallucination—supported by a validated prompt strategy and manual audits.
>
> * **FNSPID**: We aligned financial time series with news articles using custom relevance prompts and timestamp filters, producing a temporally coherent, multimodal dataset grounded in real-world trading constraints.
>
> In addition to these datasets, Time-IMM introduces **three formal metrics to quantify time series irregularity**—feature observability entropy, temporal observability entropy, and mean inter-observation interval (defined in Appendix C). These metrics provide a principled way to measure structure beyond surface-level statistics like average sampling rate, and offer the community new tools for dataset characterization.
>
> Taken together, the combination of **original data collection, rigorous multimodal construction, curated alignment strategies, and structural benchmarking metrics** makes Time-IMM more than a simple collection. Our goal is to provide a **practical, extensible, and principled foundation** for benchmarking in irregular, asynchronous, and multimodal forecasting settings—one that reflects real-world data challenges while supporting meaningful scientific progress.
>
> ---
>
> > ### W2: Text is the only secondary modality; limited applicability in multi-modal or asynchronous scenarios (e.g., images, audio)
>
> While our initial release focuses on **text** as the auxiliary modality, we agree that supporting other asynchronous data types (e.g., images, audio) is valuable for broadening applicability. As discussed in the **“Beyond Textual Modalities”** paragraph of our **Future Work** section, Time-IMM is designed as a modular foundation that encourages such extensions.
>
> To demonstrate this, we have now extended the **GDELT** dataset to include an **image modality**, by scraping the images embedded in each news article and encoding them using a **pretrained Vision Transformer (ViT)**. This required **only replacing the LLM-based text encoder with an image encoder**, without modifying the rest of the IMM-TSF pipeline—showcasing the framework’s plug-and-play design.
>
> Below, we show the forecasting results with different auxiliary modalities on GDELT:
>
> | Extra Modality | MSE (↓ better) |
> | -------------- | -------------- |
> | None           | 1.067          |
> | Text           | **1.066**      |
> | Image          | 1.675          |
>
> While the image-only variant underperforms text in this setting (likely due to weaker semantic alignment with financial time series), this experiment confirms that **IMM-TSF supports alternative modalities with minimal changes**, enabling researchers to explore cross-modal irregularity at scale. The only required modification was swapping the encoder component:
>
> ```python
> # Original text-based encoder (frozen LLM)
> notes_input ← encode_text(raw_input, model=text_encoder)
>
> # Replaced with image encoder (no other changes to fusion pipeline)
> notes_input ← encode_image(raw_input, model=image_encoder)
> ```
>
> This highlights one of IMM-TSF’s core strengths: its **modality-agnostic architecture**, which allows benchmarking, fusion, and forecasting with a wide variety of asynchronous input types. We will release this image-augmented GDELT variant as part of a follow-up and include this modality-switching example in the appendix.

---

> > ### Comment · Reviewer_4Has · 2025-08-04
> >
> > Thanks for addressing my concerns. I have adjusted my scores accordingly.

---

### Official Review · Reviewer_s2q2 · 2025-07-02

**Rating:** 6
**Confidence:** 5

**Summary:**

The paper introduces a new benchmark, Time-IMM, along with its library, IMM-TSF, for studying irregular, multimodal, multivariate time series. This work addresses an important gap in the current literature, as most existing time series benchmarks assume clean, regularly sampled, and unimodal data—a setting far removed from real-world applications in healthcare, finance, and climate science. The proposed dataset and benchmark are timely and address a highly relevant research need.

**Dataset Code Accessibility:**

Yes

**Ethical Considerations:**

No, there are no or only very minor ethics concerns

**Final Justification:**

Thanks for addressing my concerns. I am going to mantain the original scores.

**Limitations Weaknesses:**

However, I have several major concerns that should be addressed before the paper can be considered for publication:

1. Taxonomy clarity and distinction between irregularity types
The proposed taxonomy is an important contribution, but some categories and subcategories are not clearly distinguished, leading to potential overlaps and ambiguity:
 (a) The difference between "Adaptive or Reactive Sampling" and "Human-Initiated Observations" is unclear, as both involve conditionally triggered data collection. The paper would benefit from more precise definitions and concrete examples to clearly distinguish these subtypes;
(b) For constraint-based irregularities, the rationale for splitting them into three subcategories is not well justified. What modeling or analytical benefit does this finer categorization offer?
(c) Additionally, the "Missing Data / Gaps" category overlaps conceptually with other categories such as adaptive sampling, human-initiated logging, and multi-source asynchrony, all of which can result in missing values. The authors should provide a more detailed and principled explanation of how each category is defined and what modeling implications arise from these distinctions.

2. Ambiguities in dataset construction
Several points in the dataset construction pipeline require further clarification:
(a) The initial filtering step for textual data is only briefly mentioned. How are "semantically relevant" documents identified and filtered? What criteria or thresholds are used?
(b) The concept of asynchronous timestamping is central to the dataset but is insufficiently explained. Including a detailed illustrative example in the main text (not just in appendices) would significantly aid reader understanding.
(c) Figure 2 is visually appealing but not very informative as presented. It would be helpful to include annotated descriptions of what patterns the reader should observe and how they relate to the proposed taxonomy of irregularity.

3. Pre-alignment wtrategy in IMM-TSF
In the benchmark section, the manuscript mentions a pre-alignment strategy but does not explain it in sufficient depth:

a. The authors should elaborate why pre-alignment is necessary for adapting regular TS models to irregular data in this context.
b. More details should be included about the specific pre-alignment approach used (e.g., interpolation method, relative time encoding).
c. The paper should also justify the choice of this strategy over alternatives such as Gaussian Processes (GPs) or multi-task GPs, which have been proposed in recent literature (e.g., https://arxiv.org/pdf/2407.00840) for handling irregular sampling without alignment.

4. Text Encoder Size Considerations
The paper evaluates several frozen LLMs (e.g., GPT-2, BERT, LLaMA-3.1-8B, DeepSeek), but omits larger models such as LLaMA-3.1-70B or Qwen-3-32B. Do the authors consider these larger models impractical due to resource constraints, or is their use unnecessary due to task scale? It would be helpful to discuss the trade-offs between model capacity and performance, particularly in small-scale multimodal datasets where large LLMs might not yield proportional benefits.

5. Computational cost and model efficiency
While the benchmark includes a variety of forecasting and fusion modules, there is little discussion about computational efficiency:
The paper should include an analysis of the computational trade-offs between different fusion modules and text encoders (e.g., GRU-based fusion vs. cross-attention, GPT-2 vs. LLaMA). For practical deployment in resource-constrained settings (e.g., clinical or embedded systems), model size and inference time are as important as accuracy.

**Strengths Contributions:**

The paper introduces a new benchmark, Time-IMM, along with its library, IMM-TSF, for studying irregular, multimodal, multivariate time series. This work addresses an important gap in the current literature, as most existing time series benchmarks assume clean, regularly sampled, and unimodal data—a setting far removed from real-world applications in healthcare, finance, and climate science. The proposed dataset and benchmark are timely and address a highly relevant research need.

---

> ### Author Rebuttal · Authors · 2025-07-30
>
> > ### W1.1: Precise distinction between "Adaptive/Reactive Sampling" vs "Human-Initiated Observations"
>
> We agree this distinction should be clearer and will revise the text accordingly.
>
> * **Adaptive/Reactive Sampling** is triggered automatically by the system in response to environmental changes—for example, a heart monitor increasing sampling when arrhythmia is detected.
> * **Human-Initiated Observations** occur when a person manually chooses to record data—for instance, a nurse measuring vitals only when deemed necessary.
>
> The key difference is who or what decides when to record: adaptive/reactive sampling is **automatic** (machine-triggered), while human-initiated observation is **manual** (human-triggered). Both are condition-based, but the decision-making is either built into the system or made by a person.
>
> ---
>
> > ### W1.2: Why split "Constraint-based" into 3 sub-types?
>
> We split the "Constraint-based" category into three sub-types to reflect **different sources and levels of predictability** in when data becomes unavailable. While all involve externally imposed constraints, they differ in temporal structure and **modeling implications**:
>
> * **Operational-Window Sampling** – *most predictable*: Gaps are strictly clock-based (e.g., the stock market is closed at night). These patterns follow rigid schedules and can be handled with periodic masks or known business-hour calendars.
>
> * **Resource-Aware Collection** – *partly predictable*: Sampling slows or pauses due to device-level constraints like battery or bandwidth. These gaps are conditionally triggered, and can often be forecasted with auxiliary signals (e.g., energy levels) or simple “on/off” models predicting data return.
>
> * **Human Scheduling / Availability** – *least predictable*: Data is missing due to human behavior—staff shifts, sleep cycles, weekends—which introduces softer, routine-driven gaps. Models benefit from incorporating time-of-day and day-of-week cues to anticipate these patterns.
>
> Distinguishing these sub-types clarifies how each constraint leads to **different modeling assumptions and strategies**.
>
> ---
>
> > ### W1.3: Overlap between "Missing Data / Gaps" and other categories
>
> We agree that the term “Missing Data / Gaps” may be confusing, as other categories—such as adaptive sampling, human-initiated logging, and multi-source asynchrony—also involve missing values. The distinction lies in **why** the data is missing.
>
> This category refers to **unintentional gaps** caused by accidents or failures—e.g., dropped signals, network issues, or faulty sensors. In contrast, missingness in other categories is **intentional or structural**, driven by resource constraints, human decisions, or asynchronous updates.
>
> This difference has modeling implications: accidental gaps are typically treated as MCAR or MAR and handled with standard imputation. In contrast, intentional gaps are often **informative** (MNAR), and their patterns can be used as input features.
>
> To clarify this distinction, we will rename the category to **“Unplanned Missing / Accidental Gaps”** and emphasize its unintentional nature, which makes the missingness less semantically meaningful for modeling.
>
> ---
>
> > ### W2.1: Explain "semantically relevant" text filtering criteria
>
> We use a single-step prompt-based pipeline with GPT-4.1 Nano to filter and summarize only **semantically relevant** documents. Each dataset employs a domain-specific prompt that asks the model to produce a five-sentence summary **only if the input is topically relevant**; otherwise, the document is skipped. This eliminates the need for a separate relevance classifier.
>
> Examples include:
>
> * **GDELT**: filters out non-eventful or off-topic news.
> * **RepoHealth**: retains only technical issues from GitHub threads.
> * **FNSPID**: focuses on market-related news, excluding lifestyle content.
>
> To validate this pipeline, we manually reviewed 250 summaries and found **100% recall** and **88.4% precision** in identifying relevant documents.
>
> ---
>
> > ### W2.2: Add an illustrative, inline example of asynchronous timestamping
>
> We realize that we **forgot to explicitly refer to Figure 3** in the main text, which may have made our explanation of asynchronous timestamping unclear. Figure 3 in Section 3.1 (“Problem Formulation”) already illustrates the concept, showing two independent time axes—*$t_i$* for numerical observations and *$\tau_j$* for textual events—to highlight their asynchronous nature. In the revision we will add an explicit reference to Figure 3.
>
> ---
>
> > ### W2.3: Make Figure 2 explanatory
>
> We will revise Figure 2 to include annotations and captions that directly tie visual patterns to their corresponding irregularity types in the taxonomy. Key examples include:
>
> * **Adaptive / Reactive Sampling (RepoHealth)**: The plot shows dense bursts of observations during periods of high commit activity in GitHub repositories, and sparser intervals during quieter phases—demonstrating how sampling frequency adjusts based on system dynamics.
>
> * **Operational Window Sampling (FNSPID)**: The data clearly shows that observations occur only during trading hours, with consistent gaps overnight and on weekends—reflecting a hard operational constraint.
>
> * **Multi-Source Asynchrony (EPA-Air)**: Traces from different environmental sensors (e.g., temperature, humidity, air quality) appear on misaligned timelines, emphasizing the lack of synchronization between data sources.
>
> These visual patterns will be explicitly labeled in the figure and cross-referenced to their corresponding irregularity types in the taxonomy.
>
> ---
>
> > ### W3: Clarify and justify the pre-alignment strategy in IMM-TSF, including motivation, implementation details, and comparison to GP-based alternatives
>
> We provide full details of our canonical pre-alignment method in Appendix H, along with an illustrative example in Figure 7. Below is a brief explanation of why we use this strategy and how it differs from GP-based alternatives.
>
> **Why pre-alignment is needed**:
> To adapt regular forecasting models (e.g., PatchTST, Informer) to irregular data, we must convert variable-length, unevenly spaced inputs into fixed-length tensors. Rather than **interpolate** or **impute**—approaches that risk introducing **inductive bias**—we follow prior work (e.g., t-PatchGNN) and use canonical pre-alignment, which retains the original timestamps and missingness mask, allowing models to learn over irregular patterns directly.
>
> **How it works**:
> We construct a global temporal grid, then fill observed values, a binary mask, and normalized timestamps. Future query times are also appended, marked as unobserved. This creates a consistent, structured input while preserving all irregularity information.
>
> **Why not use Gaussian Processes (MGP)**?
> MGP-based methods, such as those used in MUSE-Net, perform full imputation as a preprocessing step. Our goal is to benchmark models under irregular sampling *without imputing missing values*, so that models must directly reason over missingness and timestamp gaps. Pre-alignment provides a more transparent, assumption-free interface for that purpose.
>
> We will clarify this in Section 4 and direct readers to Appendix H for implementation details.
>
> ---
>
> > ### W4: Explain omission of very large LLMs (e.g., Llama-70B)
>
> Larger models were excluded due to their high memory demands (≥48 GB) and limited added value. Due to time and compute limits, we evaluated LLaMA-70B on three datasets and found it consistently underperformed its smaller 8B counterpart, offering no clear advantage despite significantly higher cost.
>
> |Dataset|GPT-2|BERT|LLaMA-8B|LLaMA-70B|DeepSeek-7B|
> |-|-|-|-|-|-|
> |GDELT|**1.066**|**1.066**|**1.066**|1.069|1.067|
> |RepoHealth|**0.540**|0.541|0.545|0.547|0.545|
> |MIMIC|0.514|0.571|**0.491**|0.596|0.524|
>
> These results support our decision to focus on smaller-scale encoders that offer a better trade-off between efficiency and accuracy. Nonetheless, IMM-TSF is fully modular, allowing users to substitute larger models via a simple config change if desired.
>
> ---
>
> > ### W5: Computational cost
>
> ### **Fusion module efficiency**
>
> Both **TTF modules** have a fixed parameter count, while **MMF modules** scale with the number of time-series features. To illustrate concrete trade-offs, we report below the number of parameters, training time, inference time, and forecasting performance (MSE) for four fusion module combinations on the RepoHealth dataset (10 features):
>
> |TTF + MMF Module|# Params|Training Time (s)|Inference Time (ms)|MSE|
> |-|-|-|-|-|
> |TTF_RecAvg + MMF_GR_Add|1.20M|2.91|80.05|**0.540**|
> |TTF_RecAvg + MMF_XAttn_Add|4.73M|3.94|99.20|0.687|
> |TTF_T2V_XAttn + MMF_GR_Add|4.45M|4.97|131.26|0.546|
> |TTF_T2V_XAttn + MMF_XAttn_Add|7.98M|5.03|154.34|0.683|
>
> For **TTF**, both RecAvg and T2V-XAttn perform similarly, suggesting simple recency-based averaging is sufficient. For **MMF**, GR-Add outperforms XAttn-Add, likely due to its gating mechanism that filters out noisy text signals. Overall, **TTF\_RecAvg + MMF\_GR\_Add** offers the best balance of speed and accuracy, making it a strong default choice.
>
> ### **Text encoder efficiency**
>
> All textual encoder outputs are projected to a shared **768-dimensional space** via a learned linear transformation before fusion. Since the **text encoders are frozen**, we precompute the document embeddings once and reuse them during training, which significantly reduces runtime overhead.
>
> Here is a comparison of computational cost for generating embeddings across various LLMs:
>
> |Text Encoder|# Params|Embedding Time (s)|MSE|
> |-|-|-|-|
> |GPT-2|124.44M|0.04|**0.540**|
> |BERT|109.48M|**0.02**|0.541|
> |DeepSeek-7B|6,481.17M|1.02|0.545|
> |LLaMA-8B|7,504.92M|1.06|0.547|
> |LLaMA-70B|69,503.03M|17.16|0.545|
>
> While **larger models incur higher one-time costs**, all encoders remain practical in our setup due to precomputation. For real-time applications or edge deployment, **GPT-2 or BERT** offer the best efficiency-accuracy trade-off.

---

### Note · Authors · 2025-08-12

We thank the AC and reviewers for a constructive process. This note summarizes consensus and concrete changes.

---

## Reviews
One **Strong Accept**; one **raised** their score after rebuttal; two **Borderline Accept**. Trajectory is positive.

---

## Clarifications & additions
- **Taxonomy.** “Adaptive/Reactive” = system-triggered; “Human-Initiated” = human-triggered. The three **constraint-based** subtypes differ in predictability and modeling (calendar masks vs device on/off vs human-routine cues). We will **rename** “Missing Data/Gaps” to **Unplanned Missing / Accidental Gaps** to separate unintentional outages from informative, intentional sparsity.
- **Asynchrony & figures.** We will point to the asynchronous-timestamp example and annotate Fig. 2 to link visuals to taxonomy items.
- **Dataset construction.** We describe a one-step, prompt-based text filtering pipeline and report a manual audit (250 samples: 100% recall, 88.4% precision) supporting relevance and low hallucination.
- **Pre-alignment.** To adapt regular TS models without imputing away irregularity, we use a canonical pre-alignment that preserves timestamps and masks; GP-based alternatives are discussed.
- **Efficiency & model size.** We report parameter/runtime trade-offs; text embeddings are **precomputed**; larger LLMs showed no benefit in spot checks (70B underperformed 8B on three datasets), so we default to smaller frozen encoders while keeping the pipeline swappable.
- **Generalization.** Rolling-origin cross-validation shows stable performance over time.
- **Modality-agnostic design.** We show plug-and-play substitution of the text encoder with a vision encoder on GDELT.

---

## Scope
Our contribution is the **cause-driven** Time-IMM dataset suite and the **plug-and-play** IMM-TSF benchmark library for irregular multimodal forecasting, with modular design for easy swapping of encoders, fusion strategies, and modalities. Time-IMM spans nine distinct irregularity reasons and supports extensions to combined irregularities and new modalities.

---

## Camera-ready
We will integrate the taxonomy clarifications/renaming, expand figure annotations, include the added analyses/results, and polish datasheets, licenses, and fetch scripts.

---

## Take-away.
Time-IMM reframes irregular TS by **why** it is irregular and pairs that with multimodal benchmarks and baselines. With positive movement post-rebuttal, we believe it offers reproducible value to the Datasets & Benchmarks community.

---

### Decision · Program_Chairs · 2025-09-18

**Decision:**

Accept (poster)

**Comment:**

All the reviewers support the acceptance of the paper, with three Borderline Accept and one Strong Accept. All the reviewers acknowledge the contributions of this benchmark, which studies irregular, multi-modal, multi-variate time series in real-world application and supports asynchronous integration of numerical and textual data.

According to all the comments, the AC agrees with reviewers and decides to accept it, and suggests improving the full paper further in terms of reviewers’ comments in the camera-ready version.